METHODS AND RESOURCES

# Hyperspectral imaging in animal coloration research: A user-friendly pipeline for image generation, analysis, and integration with 3D modeling

**Benedict G. Hogan** [ID]*, **Mary Caswell Stoddard** [ID]*

Department of Ecology and Evolutionary Biology, Princeton University, Princeton, New Jersey, United States of America

* bhogan@princeton.edu (BGH); mstoddard@princeton.edu (MCS)

**Data Availability Statement:** All hyperspectral images and all code for this analysis are available on Dryad (DOI: 10.5061/dryad.j0zpc86nf) and GitHub (https://github.com/bghogan/

## Abstract

Hyperspectral imaging—a technique that combines the high spectral resolution of spectrophotometry with the high spatial resolution of photography—holds great promise for the study of animal coloration. However, applications of hyperspectral imaging to questions about the ecology and evolution of animal color remain relatively rare. The approach can be expensive and unwieldy, and we lack user-friendly pipelines for capturing and analyzing hyperspectral data in the context of animal color. Fortunately, costs are decreasing and hyperspectral imagers are improving, particularly in their sensitivity to wavelengths (including ultraviolet) visible to diverse animal species. To highlight the potential of hyperspectral imaging for animal coloration studies, we developed a pipeline for capturing, sampling, and analyzing hyperspectral data (here, in the 325 nm to 700 nm range) using avian museum specimens. Specifically, we used the pipeline to characterize the plumage colors of the King bird-of-paradise (*Cicinnurus regius*), Magnificent bird-of-paradise (*C. magnificus*), and their putative hybrid, the King of Holland's bird-of-paradise (*C. magnificus x C. regius*). We also combined hyperspectral data with 3D digital models to supplement hyperspectral images of each specimen with 3D shape information. Using visual system-independent methods, we found that many plumage patches on the hybrid King of Holland's bird-of-paradise are—to varying degrees—intermediate relative to those of the parent species. This was true of both pigmentary and structurally colored plumage patches. Using visual system-dependent methods, we showed that only some of the differences in plumage patches among the hybrid and its parent species would be perceivable by birds. Hyperspectral imaging is poised to become the gold standard for many animal coloration applications: comprehensive reflectance data—across the entire surface of an animal specimen—can be obtained in a matter of minutes. Our pipeline provides a practical and flexible roadmap for incorporating hyperspectral imaging into future studies of animal color.

HyperBirdOfParadise), respectively. An archived copy of the code is stored on Zenodo (DOI: 10. 5281/zenodo.13749562).

**Funding:** This work was supported by a Packard Fellowship for Science and Engineering (to MCS), an Alfred P. Sloan Research Fellowship (to MCS), a Polymaths Award from Schmidt Sciences (to MCS), National Science Foundation grant 2029538 (to MCS), the High Meadows Environmental Institute, and general funds from Princeton University (to MCS). The funders had no role in study design, data collection and analysis, decision to publish, or preparation of the manuscript.

**Competing interests:** The authors have declared that no competing interests exist.

**Abbreviations:** LWS, long wavelength-sensitive; MWS, medium wavelength-sensitive; PCA, principal component analysis; SWS, short wavelength-sensitive; VS, violet-sensitive; UMAP, uniform manifold approximation and projection.

## Introduction

Investigating animal coloration has proven useful for unraveling the evolutionary processes that generate phenotypic diversity [1]. In recent years, quantifying animal color has entailed two main methods: spectrophotometry and multispectral digital photography [2–6]. Spectrophotometry permits measurements of the reflectance across many wavelengths for a single point, while multispectral photography captures detailed spatial information in a limited number of color channels. Each approach has distinct advantages, but choosing one method over the other forces a trade-off between high spectral resolution (spectrophotometry) and high spatial resolution (photography, see Table 1).

Spectrophotometry has been the go-to tool for animal color researchers since the 1980s and 1990s, when spectrophotometers became widespread, affordable, and capable of capturing wavelengths beyond the human visible range [4,6,8]. Using this approach, reflectance measurements of animal coloration are obtained with a spectrometer and a broadband light-source connected via optical fibers to a small probe. The probe, placed on a small region of an animal's feathers, fur, or skin, captures detailed information about the light reflected over a broad range of wavelengths. The resulting reflectance spectrum has high spectral resolution (i.e., comprising hundreds of spectral measurements), but the approach suffers from a lack of spatial resolution: spectrophotometry typically provides inadequate 2D and 3D spatial information about animal color patterning. This is because only point samples can be obtained, so color sampling across the body of an animal (for example, a taxidermy specimen) is typically sparse. Consequently, researchers might fail to measure color patches (body regions with apparently uniform color; [9]) that are visible to some animals but not to humans [10] or fall short of capturing meaningful variation within a given color patch [11].

Over the last 15 years, multispectral digital photography has emerged as a complementary approach to spectrophotometry. Multispectral photography typically involves using a consumer camera that has been modified for broadband light sensitivity (to permit ultraviolet sensitivity, for example). External lens filters are sequentially attached to the camera, allowing for a series of images to be captured across multiple color channels encompassing (human) visible and ultraviolet wavelengths [4,5]. These images can then be stacked and processed (i.e., controlling for nonlinearity in raw pixel values, the spectral sensitivity of the particular model of camera, and lighting conditions) using specialized software. The result is a digital multispectral image comprising multiple channels: typically ultraviolet, blue, green, and red, or sometimes more (see [12]). Ultimately, multispectral photography generates images with high spatial resolution [4,5] because each pixel in the image represents a measurement of color. However,

**Table 1. Table highlighting some differences and similarities among spectrophotometry, multispectral photography, and hyperspectral imaging, with respect to spectral resolution, spatial resolution, sample selection, animal vision models, and cost.** Not included in the table is temporal resolution because the amount of time required to sample an individual animal is highly variable both within and among techniques. For spectrophotometry, individual measurements are collected in under a second, but typically several measurements per patch and several patches per animal are collected, resulting in a temporal resolution spanning from several seconds to minutes. For multispectral photography, capture with limited spectral sensitivity (e.g., omitting ultraviolet light) can allow for video-speed capture, but increasing spectral sensitivity typically slows capture to the range of minutes per image (but see [7] for multispectral video including ultraviolet light). For hyperspectral imaging, a similar trade-off can occur: full-frame hyperspectral imagers with limited spatial and spectral sensitivity can capture images at video-speed. The hyperspectral imager used in the current paper has high spatial and spectral resolution, but each image takes several minutes to capture.

| | Spectrophotometry | Multispectral photography | Hyperspectral imaging |
|---|---|---|---|
| Spectral resolution | High | Low | High |
| Spatial resolution | Low | High | High |
| Sample selection | Manual | Comprehensive | Comprehensive |
| May be applied to animal vision models | Yes | Yes | Yes |
| Cost | Low-medium | Low | Currently high |

each of these pixels contains only a few spectral measurements (one for each channel), severely limiting spectral resolution. Overall, multispectral photography resolves some of the challenges of spectrophotometry—it captures detailed spatial information (including patterning) and permits dense sampling across an animal's body—but at the expense of spectral resolution.

To quantify animal color accurately and comprehensively, it is desirable to capture both high spectral resolution and high spatial resolution [13]. Although reflectance spectra (high spectral resolution) and multispectral images (high spatial resolution) are both amenable to visual modeling and can be used to address a range of questions about how animals perceive color signals (reviewed in [14,15]), for many studies researchers must choose one approach in favor of the other. Reflectance spectra are essential when the goal is to identify metamers—two colors that look identical to a viewer but in fact have different spectral properties [16]. Moreover, knowing the shape and features of reflectance spectra can be especially valuable for addressing questions about the physical properties of an animal's color, such as those related to the mechanisms underlying color production [17–19] or to non-signaling functions of color like thermoregulation [20,21] and mechanical strength [22]. In these cases—in which a direct link to a particular visual system is often unimportant or unknown—analyses of reflectance spectra are usually the most appropriate (reviewed in [14]). On the other hand, for questions related to spatial information—such as those involving the distribution of patches and fine-scale patterning on an animal's body (see [9,23])—multispectral images have the clear advantage [3–5]. Multispectral photography can also be combined with photogrammetry to produce multispectral 3D models, allowing for integrative studies of color and 3D body morphology [24].

Hyperspectral imaging is a compelling alternative to spectrophotometry and multispectral photography: it offers the best of both worlds by providing both high spectral resolution and high spatial resolution (Table 1). Hyperspectral imagers work by simultaneously capturing high-resolution color information at many locations, in essence taking a photograph with hundreds of channels of color information instead of relatively few as in traditional (RGB) or multispectral photography (see Fig 1 and Fig D in S1 Appendix). A hyperspectral imaging system separates the light spectrum into a series of narrow bands, each of which corresponds to a small wavelength range (Fig 1). The end result is a data array or stack (often termed a datacube), recording radiance along two spatial dimensions (width, height) and one spectral dimension (wavelength). Each pixel in a hyperspectral image can correspond to a full reflectance spectrum (Fig 1C), and an image of the brightness of the whole specimen can be produced for any desired wavelength (by taking a slice of the data array, see Fig 1A and 1D). While hyperspectral imaging has become an important tool in agriculture, geology, medicine, archeology, art history, and human color science [16,25,26], applications of hyperspectral imaging to animal color quantification are in their relative infancy. To study animal color, early adopters of hyperspectral imaging used the technique to investigate the tuning of dichromatic visual pigment sensitivities to real forest and underwater scenes [27], surface color in butterfly wings [28], iridescent spiders [29], beetle wing cases [30], and frog skin [31], and camouflage in cuttlefish [32] and crabs on natural substrates [33]. Additionally, researchers have used hyperspectral imaging to simulate avian perception of warning colors on lepidopteran wings [34], to measure color and simulate warning coloration for newts [35], and to measure bioluminescence in sharks [36]. Kim and colleagues [37] combined hyperspectral imaging with 3D scanning to produce the first—to our knowledge—hyperspectral 3D model of an animal museum specimen, a colorful Papuan lorikeet (*Charmosyna papou goliathina*; now considered Stella's lorikeet, *Charmosyna stellae*).

Despite these applications, widespread adoption of hyperspectral imaging in animal color research has been slow. There are three main obstacles. First, hyperspectral imaging systems are expensive and require substantial computational power and memory for data processing

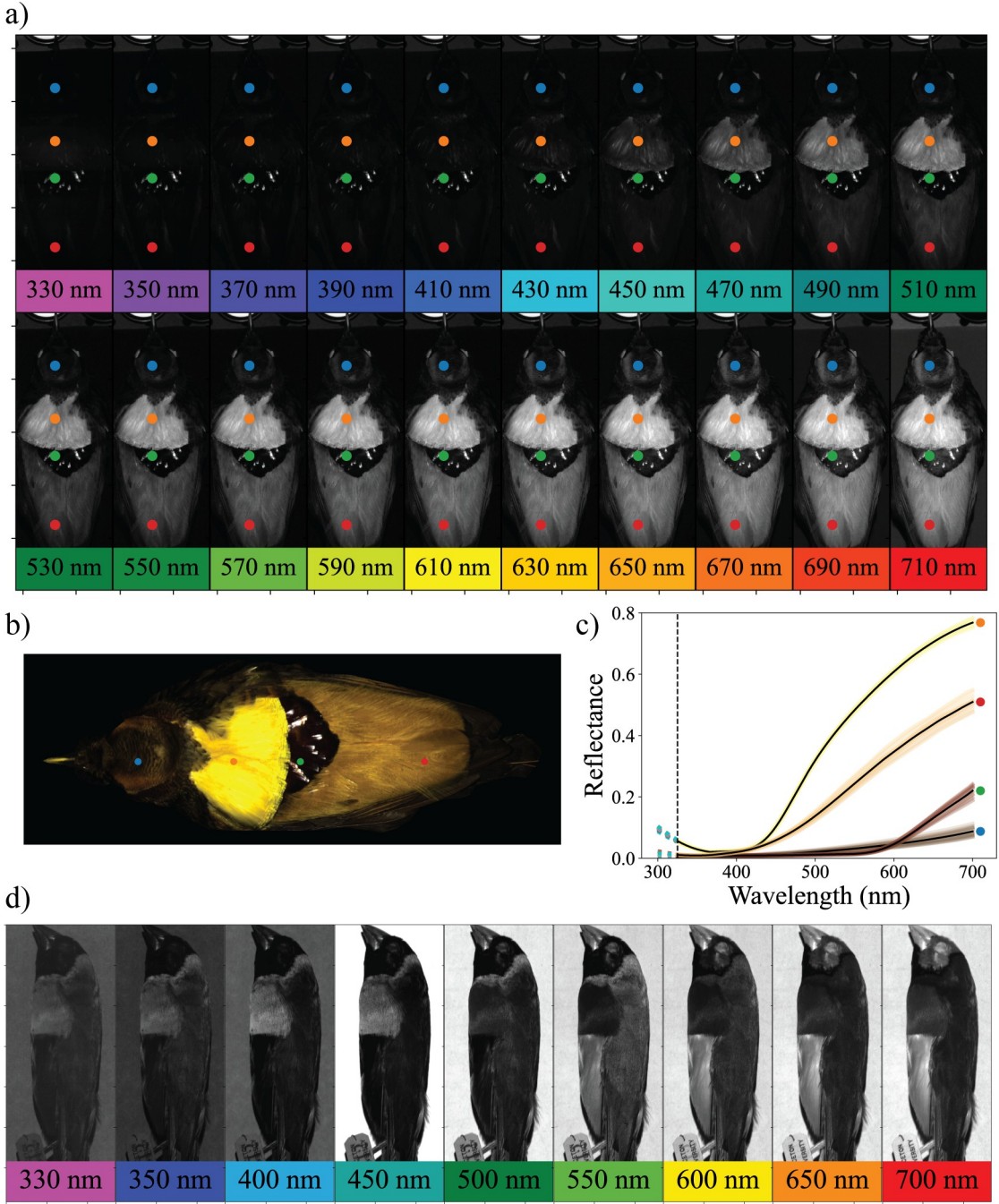

**Fig 1. Hyperspectral imaging data captures spectral and spatial information.** (a) Montage of selected slices across the spectral dimension of the hyperspectral data-cube, each indicating in grayscale the reflectance of one specimen (Magnificent bird-of-paradise dorsal view) at the wavelength indicated in the inset colored boxes. The blue, orange, green, and red dots indicate the plumage regions sampled (see also b and c). Note that intervals of 20 nm were chosen for plotting; the actual data were acquired in intervals of ~2.5 nm. (b) RGB image of the dorsal view of the specimen generated using 3 select slices (425 nm, 550 nm, 600 nm) with plumage regions sampled again indicated with colored dots. (c) Plot of the reflectance of each of the plumage samples shown in (a) and (b); each pixel in a hyperspectral image corresponds to a full reflectance spectrum. The dotted vertical line indicates the lower sensitivity bound of the hyperspectral image; reflectance values below this wavelength were extrapolated. (d) The birds-of-paradise measured in this study reflect relatively little ultraviolet light (300–400 nm). For comparison, shown here are hyperspectral images of the lateral view of a Gouldian finch (*Chloebia gouldiae*), which reflects substantial ultraviolet light in the breast patch. The data underlying this figure can be found at https://doi.org/10.5061/dryad.j0zpc86nf [99].

[37]. Second, hyperspectral imagers that are sensitive to the entire portion of the visible spectrum relevant to animals have not typically been commercially available [38]; for many animals, this range is 300 nm to 700 nm [39], with 300 nm to 400 nm comprising ultraviolet light. Finally, we lack a strong conceptual framework—and corresponding software tools—for analyzing hyperspectral data in a way that is convenient for animal color research. Fortunately, the first two of these challenges are quickly fading. Hyperspectral imaging systems are becoming increasingly affordable and computationally practical, and some imagers now have sensitivity across wavelengths (including ultraviolet) relevant to animal visual systems [34]. Thus, the stage is set for wider uptake of hyperspectral imaging, which could be a game-changing tool in animal color research. All of the advantages to multispectral photography [2,5] apply to hyperspectral images, but with the additional benefit of high-resolution spectral data. For example, a single data set of hyperspectral images might now be used in two ways: (i) to ask questions about an animal's coloration and patterning (i.e., the images can be processed using models of animal vision and analyzed using existing tools for multispectral images); and (ii) to ask questions about the physical properties of the colors themselves (i.e., by comparing the shape and features of the reflectance spectra to those of known pigments and structural colors).

We currently lack a user-friendly, step-by-step pipeline for obtaining and analyzing hyperspectral images for animal color research. To address this gap, we developed such a pipeline and demonstrated how it can be used to visualize and compare animal colors in hyperspectral images. We used a commercially available Resonon Pika NUV imager (Resonon, Montana, United States of America), but the pipeline is general to hyperspectral animal coloration data captured by any hyperspectral imager. Our pipeline includes: imaging, sampling reflectance spectra from the images, and embedding the resulting colors into receiver-independent and receiver-dependent color spaces. To illustrate how hyperspectral imaging can provide new, detailed insights about animal color, we apply our pipeline to a study of the plumage of a rare hybrid bird-of-paradise, with only around 25 male museum specimens known in the world [40]. Birds, the most colorful land vertebrates, are especially amenable to hyperspectral imaging and analysis. Their colors are produced by diverse mechanisms (pigments and structural colors), they have tetrachromatic color vision, and bird specimens are readily available in museum collections [3]. Overall, hyperspectral imaging may represent a new gold standard for many avenues of animal coloration research.

## Study system: Birds-of-paradise

The birds-of-paradise (family: Paradisaeidae), native to New Guinea and Australia, are a diverse and charismatic group known for their elaborate multimodal courtship displays [41]. The family comprises 44 species in 16 or 17 genera [40,42] (Fig 2). Different species exhibit a strong tendency to hybridize, with hybridization known or suspected among very different-looking species [43,44]. One rare bird-of-paradise hybrid—with just ~25 male museum specimens in the world [40]—is the King of Holland's bird-of-paradise (*Cicinnurus magnificus x C. regius*) [45–48]. Initially thought to be a new species (by Meyer in 1875 [47]), the King of Holland's bird-of-paradise was later described as a hybrid of the Magnificent bird-of-paradise (*C. magnificus*, following the convention used in [42]; also known as *Diphyllodes magnificus*) and the King bird-of-paradise (*C. regius*). In 1927, Berlioz [45] described the King of Holland's bird-of-paradise as the perfect combination of its parents' phenotypes, writing (in French) that "an artist who tried to imagine a hybrid could not compose a more faithful reproduction." More recently, Thörn and colleagues [49] confirmed genetically that a male King of Holland's bird-of-paradise specimen was indeed an F1 hybrid. However, they also found that a male

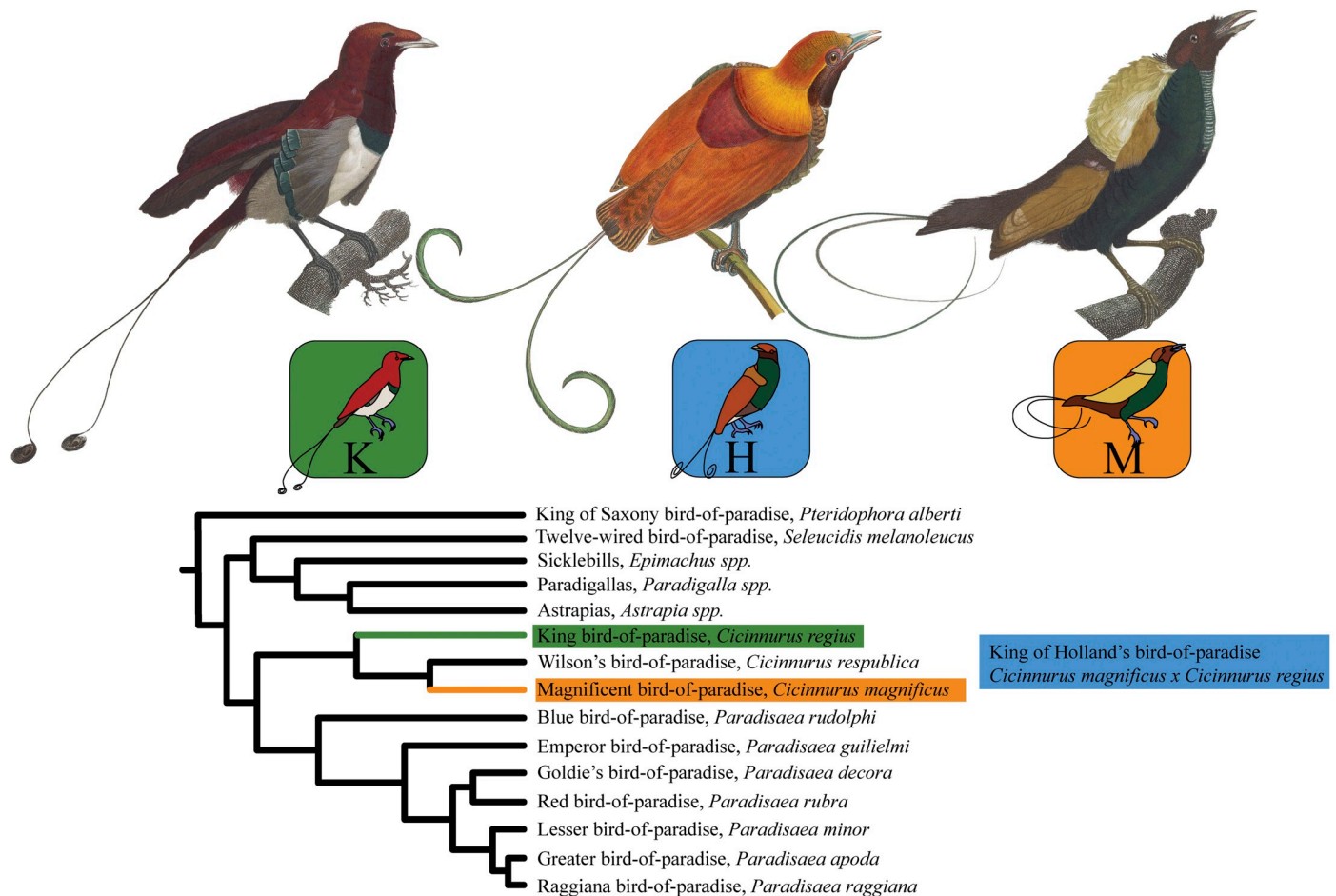

**Fig 2. The birds-of-paradise examined in this study.** Top row: Illustrations of the King bird-of-paradise, the hybrid King of Holland's bird-of-paradise, and the Magnificent bird-of-paradise. The King and Magnificent illustrations are reproduced from Levaillant and Barraband (1806, [50]), which is in the public domain. The hybrid illustration is reproduced from Gould and colleagues (1875, [51]), which is in the public domain. Middle row: Icons and colors representing the parent species and hybrid throughout the paper. Bottom row: Phylogenetic reconstruction of the subfamily Paradisaeinae, based on our replication of the phylogeny by Irestedt and colleagues [52]. All individual specimens and all genera other than *Cicinnurus* and *Paradisaea* are collapsed for clarity. See Table B and Fig J in S1 Appendix for details of the phylogenetic reconstruction.

specimen originally identified as a King of Holland's bird-of-paradise was genetically closer to a Magnificent bird-of-paradise, while a female specimen thought to be a King of Holland's bird-of-paradise was more similar to a King bird-of-paradise.

The male King, Magnificent, and King of Holland's birds-of-paradise exhibit plumage colors produced by diverse plumage mechanisms. Avian plumage coloration is generated through two broad mechanisms, pigmentary and structural color, or a combination of the two [19]. Pigmentary color is generated by the deposition of colorful substances within the feather, whereas structural color arises from nano-scale physical structures in the feather barbs and barbules. Pigmentary mechanisms commonly generate a range of plumage colors, including gray and black (melanin), brown and rufous (pheomelanin), and yellow, orange, and red (carotenoids). Structural mechanisms are typically responsible for blue and iridescent colors,

and green can be generated by a combination of yellow carotenoid pigmentation and structural blue. White plumage typically results from lack of pigmentation. Although there are few descriptions of color-production mechanisms in the King, Magnificent, and King of Holland's birds-of-paradise (but see [53]), pigmentary [54] and structural colors have been described in a range of other bird-of-paradise species [55–57]. Females of both the Magnificent and King birds-of-paradise are less colorful, with a brown head, shoulders, back and wings, and barred undersides [40]. The plumage of the female hybrid King of Holland's bird-of-paradise is likely to be drab; at least one specimen labeled as a female hybrid was recently found to be a King bird-of-paradise, though the existence of female hybrids is evidenced by at least one apparent backcross between a fertile female King of Holland's bird-of-paradise and a male King bird-of-paradise [49].

How hybridization affects coloration is not well understood, especially where coloration is non-uniform (i.e., consisting of distinct patches) across an animal's body. In birds, it is common for even adjacent patches of coloration to be generated by different structural or pigmentary mechanisms [19]. In some species, a hybrid may have a mosaic of color patches that individually match one or the other parent species [58,59], or each patch may in itself appear to be intermediate to the parent species' patches [60,61], or both [62,63]. In other cases, combinations of parental genotypes can produce apparently novel phenotypes, as in the crown patch of hybrid offspring of snow-capped (*Lepidothrix nattereri*) and opal-crowned manakins (*L. iris* [64]), and in hummingbird throat patches [65]. To what extent are the hybrid King of Holland's bird-of-paradise plumage patch colors intermediate composites of its parent species? Are individual patches intermediate, or is the hybrid a mosaic of the patches of its parents? Are the colors intermediate in terms of physical properties (reflectance spectra)? Would they be perceived as intermediate by a relevant signal receiver, such as another bird-of-paradise? To determine—quantitatively, using objective color measurements—whether the hybrid plumage patches are intermediate, refined spectral and spatial color quantification is required. This is therefore an ideal system for the application and testing of hyperspectral imaging.

## Methods

### A pipeline for obtaining and analyzing hyperspectral color data

We developed a pipeline for color analysis that takes advantage of the spectrally and spatially rich data produced by hyperspectral images (Fig 1). We provide a subset of the images, code, and data used in this pipeline so that researchers can replicate our approach and modify it for their specific research purposes. The pipeline consists of three steps: (i) imaging; (ii) sampling; and (iii) low-dimensional embedding (see Fig 3). Imaging involves acquiring images from the hyperspectral camera (Fig 3A) using SpectrononPro software (Resonon, Montana, USA). The subsequent sampling and embedding steps are performed using custom python code (available at https://github.com/bghogan/HyperBirdOfParadise). The code uses functions from several packages, including colour-science (https://github.com/colour-science/colour), spectral python (https://www.spectralpython.net/), and UMAP (https://github.com/lmcinnes/umap). Sampling involves selecting relevant colors from the hyperspectral images for analysis (Fig 3B–3D). Low-dimensional embedding involves generating informative transformations of the spectral data for interpretation and analysis (Fig 3E and 3F). Following low-dimensional embedding, researchers can perform a range of analyses to quantify color diversity and color difference. We discuss two potential low-dimensional embeddings: a visual system-independent approach and a visual system-dependent approach, here using an avian tetrahedral color space [66–68] to quantify aspects of avian color perception. As an optional fourth step of the pipeline, we describe ways in which researchers might supplement hyperspectral data with 3D models to understand how the 3D surface of the subject in the image might influence color.

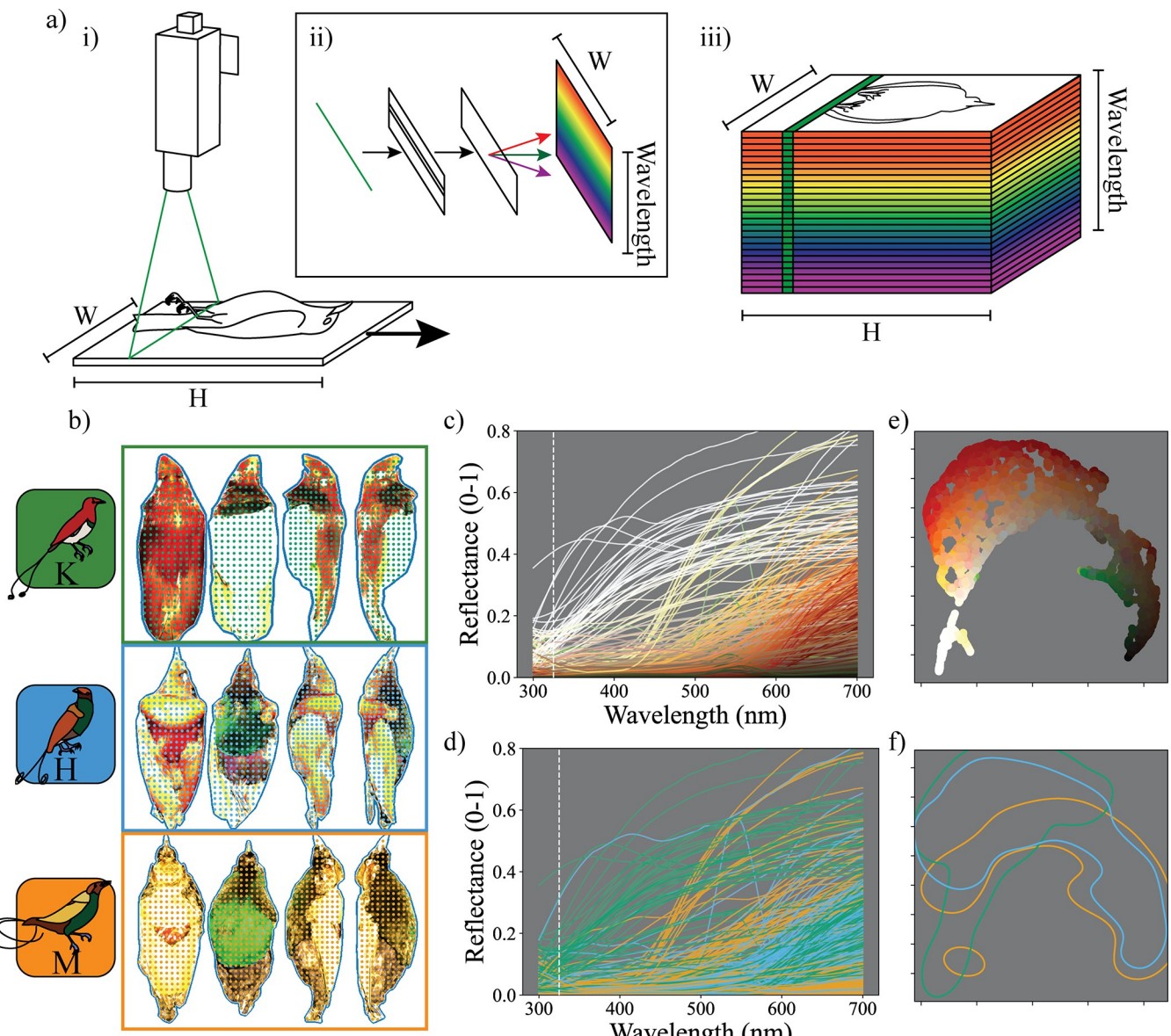

**Fig 3. Overview of the pipeline used to compare colors from the King, hybrid King of Holland's, and Magnificent birds-of-paradise.** (a) *Imaging*. (i) A diagram of the hyperspectral imager in use. A bird specimen on a linear translation stage is moved across the single-line field-of-view of the hyperspectral imager (in green). (ii) Inset diagram indicating how light from the field-of-view of the camera is processed inside the spectrometer. Incident light (in green) is split into its constituent wavelengths by a diffraction grating and brightness recorded on a sensor. (iii) A diagram of a hyperspectral data-cube being generated. The width (W) and spectral resolution (Wavelength) of the resulting image are set by the resolution of the imager and software settings, but the height (H) of the image is set by the number of exposures selected by the user. (b) *Sampling*. Six specimens (3 shown) were imaged from a number of positions (4 shown). Sampling locations from each image for the "whole" patch are indicated by colored dots. These samples correspond to reflectance spectra shown in both (c) and (d). (c) Each sampled spectrum colored according to their appearance (to human eyes, brightened by 300% for plotting). (d) The same sampled spectra shown in (c), now colored according to the species from which the sample came, as shown in the key on the far left (Green: King bird-of-paradise, Blue: hybrid King of Holland's bird-of-paradise, Orange: Magnificent bird-of-paradise). In both (c) and (d), the vertical dotted line indicates the lower bound of the hyperspectral imager's sensitivity; reflectance values below this were extrapolated. (e) *Embedding*. These same spectral samples embedded in a visual system-independent color space. We generated PCA coordinates for the samples (20 dimensions, see main text and Methods), followed by dimensionality reduction using UMAP (see main text and Methods). Embedded points are colored according to their appearance (to human eyes, brightened by 300% for plotting). (f) Kernel density plot over the embedding shown in (e), with contour lines to indicate the (80% density) contribution of each species to the distribution of colors in the color space. Line color indicates species, as shown in the key to the left of (b). The data underlying this figure can be found at https://doi.org/10.5061/dryad.j0zpc86nf [99]. PCA, principal component analysis; UMAP, uniform manifold approximation and projection.

Here, we employ this pipeline to describe the plumage coloration of two parent species (the Magnificent bird-of-paradise and the King bird-of-paradise) and their hybrid offspring (the King of Holland's bird-of-paradise). Specifically, we explore the extent to which the hybrid color phenotype is intermediate to the parental phenotypes.

**Step 1: Capturing hyperspectral images.** The first step of the pipeline is capturing hyperspectral images. We collected hyperspectral images of King, Magnificent, and hybrid specimens (two male specimens each) in the following positions: ventral, dorsal, lateral (left flank), and lateral (right flank). In these images, the specimen's body was parallel to the imaging plane. Additionally, we obtained a second image of the ventral position, this time with the specimen's head angled approximately 30˚ toward the imager, for a total of five images of the body of each specimen. We also took hyperspectral images of the dorsal side of the tail of each specimen. Specimens were loaned by the American Museum of Natural History (New York, USA, Table A in S1 Appendix). We used a Resonon Pika NUV hyperspectral camera with a CoastalOpt UV-VIS-IR 60 mm lens (Jenoptik AG., Jena, Germany). Data collection and initial spatial calibration were conducted using SpectrononPro software (Resonon). We illuminated specimens in these images using an ultraviolet-rich and spectrally broadband light source (450W Xenon ozone free arc lamp: Newport Corp., Irvine, California, powered with a Newport OPS-A1000 power supply). This light source was housed in an enclosure (Newport 66921 arc lamp housing), the output of which was first filtered for infrared light by passing through a water filter filled with deionized water (Newport 6123NS liquid filter) and then directed vertically downward using an ultraviolet-reflective turning mirror (beam turning mirror: Newport 66246, beam turning mirror housing: Newport 66225). The light, enclosure, filter, and turning mirror were placed on a small rolling table so that the turning mirror could be placed as close to the lens of the hyperspectral imager as possible. This resulted in bright directional light, from approximately the same angle as the imager's receptive field, cast over the entirety of the push-broom field of view of the hyperspectral imager (see Fig A in S1 Appendix). One limitation of this lighting set-up is that objects closer to the light source will appear artificially lighter. Therefore, improving the control and directionality of incident light is a goal for future work (see Discussion). Each image was calibrated spatially (i.e., we corrected for non-uniformity in lighting across the field of view of the imager) using SpectrononPro. This process involves collecting a "bright image" of a rectangular block of white PTFE that encompasses the entire field of view of the imager, which is used to account for differences in brightness across the field of view due to directional lighting. Each image also contained a Spectralon 99% reflectance standard (LabSphere, North Sutton, New Hampshire), which was used later to obtain spectrally calibrated reflectance images (see below, Step 2). All images were taken with the imager at the same height (90 cm) from the linear translation stage, so the scale is identical across images.

The Pika NUV camera works in a push-broom manner (Fig 3A), where a given exposure of the sensor generates one row of the data-cube at a time, in essence taking a line of 1,500 (Fig 3A, width) spectrophotometric readings simultaneously. By passing the subject across the field of view of the camera and capturing sequential exposures, the second spatial dimension of readings (Fig 3A, height) is generated. A linear translation stage is used to push the subject across the field of view at a controlled slow pace. We used SpectrononPro to operate the camera and stage, and to collect the data-cube. We collected exposures at 11 frames per second (each image took around 5.5 min to complete), and the average number of exposures was 3,600, resulting in images with a width of 1,500 pixels and an average height of 3,600 pixels, with a spectral dimension of 294 bands between 274 nm and 812 nm (each band is approximately 1.83 nm wide). Visual inspection of the data indicates that bands below 325 nm contain only noise, which means that in our configuration, this system is sensitive from 325 nm to 700 nm, encompassing almost the complete range of avian visual sensitivity (300 nm to 700 nm).

These data-cubes were saved in a standard band-interleaved-by-line format, along with a header file which contains metadata. The data-cubes produced by SpectrononPro software averaged 3.2 gigabytes in size.

Hyperspectral data-cubes can be analyzed in a number of software packages/environments, including SpectrononPro, MATLAB (MathWorks, Natick, Massachusetts), Python (Python Software Foundation, Wilmington, DE; available at www.python.org), and ENVI (NV5 Geospatial Solutions, Broomfield, Colorado). In the current study, we used custom code to extract and further analyze sets of spectra from the hyperspectral images in Python (v3.7.4, see all subsequent steps of the pipeline), but one of the key advantages of hyperspectral imaging is that the data are amenable to both image-based and spectra-based color analyses. For example, it is possible to apply visual models to entire hyperspectral images to generate cone stimulation images that can then be analyzed using popular packages, including MICA/QCPA for ImageJ [2,5,69]. We include in our code an example of this. Alternatively, users may be interested only in one or a small number of wavelength bands of interest. These may be easily extracted and analyzed as individual images. Yet another approach might be to extract only a few spectra for further color analysis using the R package pavo [70]. We include in our code examples of this approach (and see section "Some King of Holland's bird-of-paradise color patches that are spectrally intermediate would be indistinguishable from those of its parent species when modeled with avian color vision using only one spectrum per patch").

**Step 2: Extracting sample reflectance spectra from hyperspectral images.** The second step of the pipeline is to define a spatial sampling scheme in which we select certain pixels (each corresponding to a reflectance spectrum) for further analysis. Sampling from the hyperspectral images allowed us to (i) control the computational complexity of the analysis; and (ii) reduce redundancy due to spatial autocorrelation (i.e., pixels tend to share the color properties of neighboring pixels). We first manually generated for each image a set of regions-of-interest indicating the locations of pixels within select body parts/patches: whole (the entire body), breast (the pectoral shield as per [71]), belly/vent (all of the ventral side of the bird, below the pectoral shield excluding the legs), shoulder, and back (including wings, see Figs K–O in S1 Appendix for details of the placement of these patches). To sample colors from each of these patches, we defined a uniform grid of sampling locations in each image, the spacing of which was chosen to approximately equalize the number of sampling locations in images of each patch within a specimen, and across all specimens (Fig 3B). This ensured that no species, specimen, image, or patch should be overrepresented in our analysis. Because the ventral view (with the bird's long axis oriented with the image plane) did not well capture the iridescent breast patch coloration, for this patch we only sampled from the images obtained in the ventral angled position (with the bird rotated with its beak approximately 30˚ toward the camera, see Step 4 below, Discussion). Care should be taken in generating a sampling scheme; here we chose to equalize sampling across patch and specimen (so that each patch/specimen has the same number of samples), but this came at the cost of not having a uniform sampling density (samples per cm$^2$, for instance), which may be more important for other research questions. In addition, we selected samples using a regular grid. This worked well for our purposes because the bird specimens did not contain patches with complex patterning (such as spots or stripes), but care should be taken when sampling in order to collect a fair representation of the colors present.

At each sampling location, we selected a square of 10 by 10 pixels and obtained the median of the spectra corresponding to these pixels as our sample for analysis (hereafter sample reflectance spectra, Fig 3C and 3D). Each of these sample reflectance spectra was spectrally calibrated (by division) with the median spectrum of the pixels corresponding to the Spectralon 99% reflectance standard in each image. We then interpolated spectra to 1 nm bands (from

325 nm to 700 nm) and smoothed the spectra by application of a Savitzky–Golay filter (window 45 nm, polynomial order 2). It is possible that interpolating spectra to a spectral resolution of greater than 1 nm would reduce computational complexity of the analyses, potentially at little cost. We note that the values of these smoothing parameters were chosen here to aid in stable extrapolation of reflectance values below 325 nm (where small amounts of noise around 325 nm produce large differences in the resulting extrapolation), but that the result is relatively extreme smoothing (see Figs 1, 3, and B and C in S1 Appendix). Future research should determine levels of smoothing appropriate for varied applications. We then extrapolated reflectance values in the wavelength range 300 to 325 nm by applying a cubic 1D monotonic smoothing spline fit to the range 325 nm to 700 nm (see Fig B in S1 Appendix for a figure contrasting "raw" sampled spectra with filtered and extrapolated versions, and see Fig C in S1 Appendix for a figure contrasting "raw" natural spectra with filtered and extrapolated versions). Because birds have limited sensitivity to light at wavelengths shorter than 325 nm, we anticipate that results for visual system-dependent analyses will be very robust to extrapolation. The end result is that for each patch of each specimen, we collected approximately 1,000 sample reflectance spectra for analysis in Step 3. At this point, a range of analyses on the sampled reflectance spectra can be undertaken (see, for example, [8]).

**Step 3: Embedding sample reflectance spectra in low-dimensional spaces.** The third step of the pipeline is to embed the sample reflectance spectra generated in Step 2 in a low-dimensional space (Fig 3E and 3F). The advantage to this is ease of visualization; it also serves as a gateway to other analyses [8,68]. Here, we demonstrate two different ways in which to embed the reflectance spectra: in a visual system-independent color space and in a visual system-dependent space. A visual system-independent approach takes full advantage of the spectral resolution provided by the hyperspectral images because it does not *a priori* select a subset of wavelengths of interest, instead using the variability in the data to inform the embedding (Fig 3E and 3F). It also makes no assumptions about a potential signal viewer. Because this approach requires embedding high-resolution reflectance spectra, it would not be possible to produce an identical color space using multispectral photography. A visual system-dependent approach allows for spectra to be represented in a way that is relevant to a signal receiver (reviewed in [14,15]). Here, we embedded spectra in a tetrahedral color space [17,66–68]. The result of this embedding is similar but not identical to that obtained using multispectral photography (see details below).

**Embedding in a visual system-independent color space.** To embed the spectra in a low-dimensional, visual system-independent color space, we begin by applying principal component analysis (PCA, implemented using scikit-learn [72]) to our sampled spectra (one PCA per plumage patch). We subtracted the mean reflectance from each spectrum before applying PCA (see [8]). The calculated PCA coordinates (20 dimensions) are not easily interpretable since any 2D plot (for instance, axis 1 versus axis 2) may not fully capture the distances between the embedded colors. To reduce this dimensionality, we used uniform manifold approximation and projection (UMAP: see below and Fig E and F in S1 Appendix) in order to embed spectra in a lower dimensional space that attempts to maintain distances between PCA coordinates. Spectra that fall close together in this space will tend to be those with low distances between PCA coordinates, and those far apart will tend to have high distances. The result is a single, easily interpretable 2D plot of the distances among sampled spectra. Our goal here is to illustrate and explore the spectra rather than to conduct statistical analyses, but PCA coordinates are amenable to statistical approaches: for instance, one might look for statistical separation in the distributions of points of each species (e.g., using MANOVA).

**Embedding in a visual system-dependent color space.** It is often advantageous to estimate how an animal would perceive color signals. One way to accomplish this is to quantify

cone catch values (i.e., an estimate of how an animal's photoreceptors would be stimulated by a certain color/reflectance spectrum; for a review of these methods, see [14,15]). Here, to estimate how a bird-of-paradise might perceive the plumage colors of the King, Magnificent, and hybrid King of Holland's birds-of-paradise, we embedded each of our sampled reflectance spectra in a tetrahedral model of avian vision [67,68] assuming ideal illuminant and background. Birds-of-paradise are presumed to have violet-sensitive (VS) visual systems [41], so we used visual pigment sensitivities for an average VS bird (extracted from the R package pavo [70]), as well as cone ratio values of (1,1,1:2) reported for Rock dove (*Columba livia*) [73] and a chromatic Weber fraction of 0.1.

For each sample reflectance spectrum, this analysis generates 4 cone catch values, one each for a bird-of-paradise's violet-sensitive (VS), short (SWS), medium (MWS), and long (LWS) wavelength-sensitive cones. These values can be converted to 3D x, y, and z coordinates in tetrahedral color space for plotting and analysis [68]. The process here is identical to that used if a single reflectance spectrum is measured (as with a spectrophotometer); here, we have many more spectra because hyperspectral imaging allows for much more dense sample selection than would be practical with spectrophotometry. Representing reflectance data in a visual system-dependent color space is also possible with multispectral imaging, where specialized software has been developed to convert multispectral images into cone stimulation images [2,5]. However, the accuracy of data derived from hyperspectral images will be greater; multispectral imaging requires the user to make some assumptions when modeling the mapping between the sensitivity of the camera and the sensitivity of the visual system, but hyperspectral data provides (near) complete spectral information, so fewer assumptions are required.

**Step 4: Combining hyperspectral data with a 3D digital model.** The fourth step of the pipeline is optional, allowing users to generate a 3D digital model of the imaged specimen. We generated 3D models of each of our specimens using photogrammetric software (Metashape v2.0.2, Agisoft LLC, Saint Petersburg, Russia). We note that several free and open-source alternatives for photogrammetric software exist. 3D models are powerful because they combine color and morphometric data—and they are likely to become an increasingly common resource for animal color researchers [24,37]. Our specific aims for this step were twofold: first, to create a 3D model to accurately represent the specimen's shape, and second, to estimate the angle (relative to the hyperspectral imager) of the surface of the bird specimen (hereafter surface normal) for any reflectance spectrum sampled from the hyperspectral images (Fig 3B). Angular information is useful because bird plumage is not Lambertian (i.e., reflectance is a function of viewing and lighting geometry) for both iridescent [74] and probably many non-iridescent colors [75]. Supplementing our 2D image with 3D information about the surface normal will allow us to inspect the dependence of measured color on 3D angle, at the level of the patch.

During imaging, we were cautious when manipulating the specimens due to their rarity and age. We custom designed and 3D printed small stands to hold the specimen, which were then affixed to a small rotating stage, allowing us to photograph the specimen from many angles (see Fig R in S1 Appendix). For each specimen, we collected two sets of images: one with the bird supine (lying on its back) and another with the bird prone (lying on its front). For each set of images, we collected approximately 120 photographs using a consumer Sony (Minato City, Tokyo, Japan) A7 DSLR camera under typical laboratory fluorescent lighting. We rotated the bird about 360 degrees in increments of 30 degrees, resulting in 12 photographs per rotation. Between rotations, we changed the elevation and angle of the camera to capture 10 such rings of 12 photographs (see Fig S in S1 Appendix). In some cases, we collected extra photographs (largest number 158) and these were also included in the analysis.

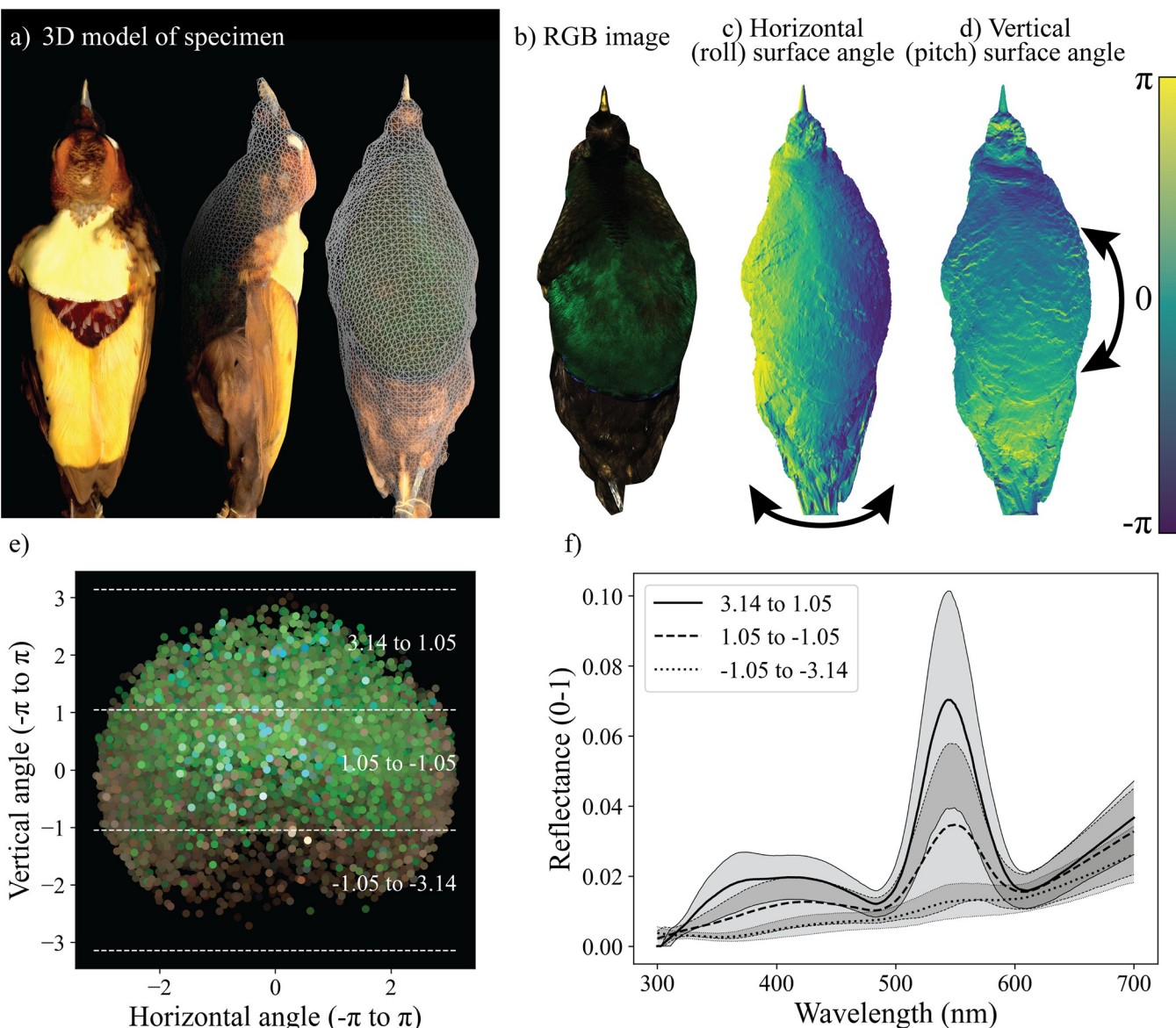

**Fig 4. Combining hyperspectral data with 3D digital models.** (a) Three renders (digital images) of one 3D model produced using our pipeline. The left image shows a color render of the 3D model. For the middle and right images, a simplified mesh is superimposed over the colored render of the model to illustrate the 3D information contained. (b) An RGB version of the ventral hyperspectral image of a Magnificent bird-of-paradise specimen. For each such hyperspectral image, we extracted 3D information from the 3D models to augment the color data. (c, d) Images illustrating the 3D information extracted from 3D models for each of our hyperspectral images. These images indicate the surface normal of the specimen relative to the camera in color from π to -π. The color indicates the horizontal component (roll, c) and vertical component (pitch, d) of the specimens' surface normal relative to the camera, with values close to 0 signifying surfaces nearly parallel to the camera, and more extreme values indicating extreme angles relative to the camera. For roll (c), π indicates that the surface normal of the specimen is facing to the left of the camera and -π indicates a normal facing to the right. For pitch (d), π indicates a surface normal facing up relative to the camera, and -π indicates a surface normal facing downwards. We generated similar information for all hyperspectral images in our data set. (e) Scatterplot of samples of the breast patch across all specimens, where the x- and y-axes represent the horizontal and vertical components of the surface normal extracted from 3D the models for each sample. Points are colored according to their appearance to human eyes (brightened by 300% for plotting). Dotted lines across the y-axis show the bins of vertical angles plotted in (f). (f) A plot of the median (black lines) and median ± the median absolute deviation (shaded areas, with overlap in shaded areas indicated by darker shading) of the reflectance of breast patch samples across all specimens. The three lines represent samples binned by the vertical (pitch) component of the samples (see E). The data underlying this figure can be found at https://doi.org/10.5061/dryad.j0zpc86nf [99].

To generate 3D structure from multiple photographs, the position and angle of each of the photographs must be estimated from the images. Metashape calls this "alignment" and computes the alignment by correlating the position of features across images (see Fig S in S1 Appendix). To facilitate this process, we generated binary masks in the software that eliminated the stationary background and other unwanted features from the alignment process. With a set of aligned cameras, Metashape can generate a triangulated 3D model, which may then be textured by algorithmically extracting colors from the photographs that generated the model and placing these on the polygonal faces of the model (see Fig T in S1 Appendix and S1 Video). We followed this process for each specimen.

In order to relate information from the 3D models to the hyperspectral images, some registration between the 2 was required (see Fig 4). We imported RGB versions of our hyperspectral images into the photograph alignments generated above. We then marked on each hyperspectral image and each 3D model a set of points of known location (for example, the tip of the beak, the lores, and chin). This allowed us to align our hyperspectral image with the 3D model (treating the hyperspectral image as if it were another source photograph), making it possible to render the 3D model from the approximate point of view of the hyperspectral camera. We rendered images of the 3D model false-colored according to the angular surface normal in relation to the camera (see Fig 4C and 4D, and S1 Video).

## Dryad DOI

https://doi.org/10.5061/dryad.j0zpc86nf [99]

## Results

### Application of the pipeline to the birds-of-paradise

Here, in a single hyperspectral data-cube, we captured high-resolution spectral information: for every pixel in the data-cube, we recovered complete, high-resolution reflectance spectra from 325 nm to 700 nm. We simultaneously collected spatial information: at each band from 325 nm to 700 nm (sampling every ~2.5 nm), we obtained a complete photograph/image of each specimen. We have therefore illustrated the utility of hyperspectral imaging as a tool to simultaneously produce data with high spectral and high spatial resolution. In the sections below, we summarize our key findings about the plumage colors of the King, Magnificent, and hybrid King of Holland's birds-of-paradise.

**(i) Analyzing plumage colors with a visual system-independent approach.** Fig 5 shows a visual system-independent embedding of colors sampled from patches of the King, Magnificent, and hybrid King of Holland's birds-of-paradise. Shown are low-dimensional embeddings of the PCA coordinates of the sampled spectra. The top row of Fig 5A–5E illustrates the distribution of colors of each patch across King, Magnificent, and hybrid specimens. The contribution of each species to the color distribution for each patch can be seen in the middle row (Fig 5F–5J). Here, approximately 80% of a species' contribution is enclosed by a density contour with color indicating King, Magnificent, or hybrid. The spectra corresponding to the most typical colors for the King, Magnificent, and hybrid King of Holland's patches are plotted in the bottom row (Fig 5K–5N).

*The hybrid King of Holland's bird-of-paradise has plumage colors that are generally intermediate between its parent species, but it more closely resembles one or the other parent in different patches.* Collectively, the King, Magnificent, and hybrid King of Holland's birds-of-paradise contain plumage colors that are red, orange, brown, yellow, green, and white (Fig 5A). In general, the colors measured in this study reflected very low levels of ultraviolet light (Fig 5K–5N). Examining the distribution of colors for the King, Magnificent, and hybrid King

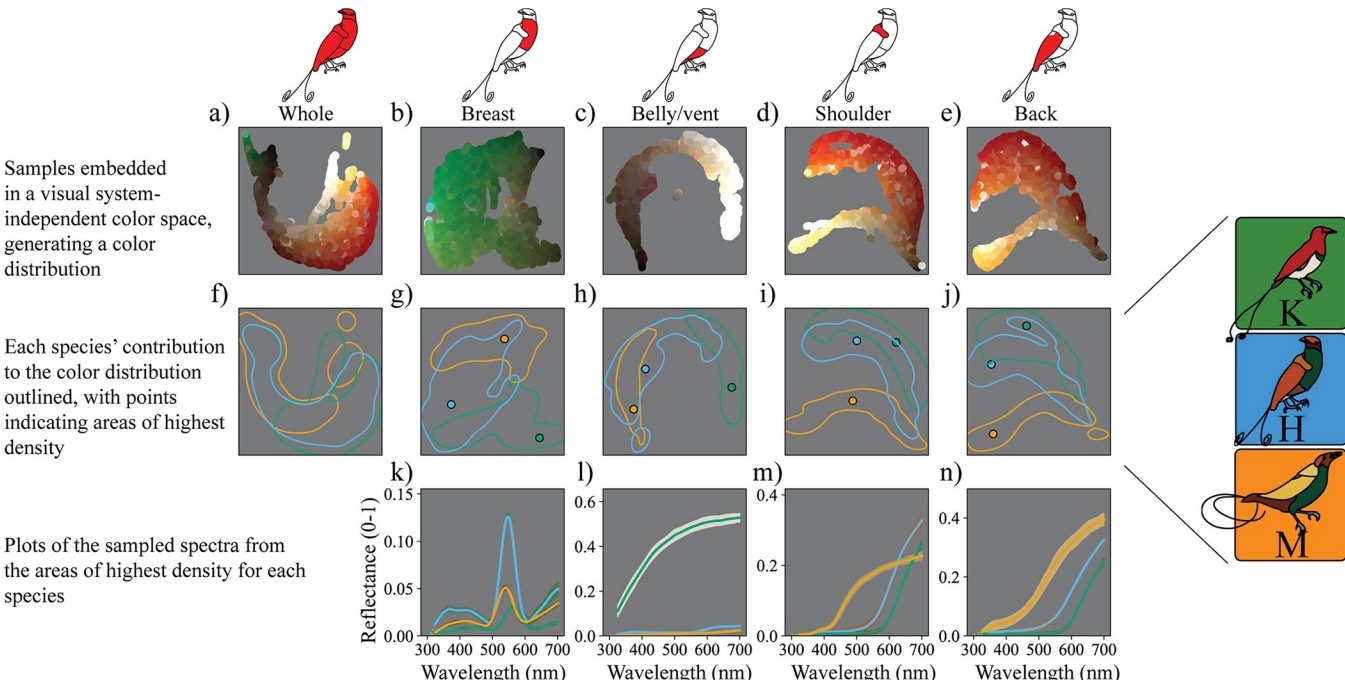

**Fig 5. Visual system-independent embedding of sampled spectra for each patch.** Top row: Icons indicate the patch represented in each column of the figure. (a–e) Plot of low-dimensional embedding (UMAP embedding of PCA coordinates) of the sample reflectance spectra, colored according to their appearance to human eyes (brightened by 300% for plotting). Note that the embeddings were computed separately for each patch; each column shows a different color space, so they should not be directly compared. (f–j) Kernel density contour plots indicating the 80% density contribution of each species to the color spaces in a–e. Line and point colors correspond to the different species as indicated in the key at the right of the figure. Areas of overlap indicate that the colors from more than one species occupy the enclosed area of color space. Points indicate the area of highest kernel density for each of the species: these areas represent the most common spectra for each species for that patch. (k–n) Plot of the 50 closest (in embedded space) spectra to the areas of highest kernel density for each patch and species (points in f–j). Bold line indicates median colored by species, shaded areas indicate median ± the median absolute deviation, and shaded colors represent approximate color of the samples (to human eyes). In general, the plumage colors measured in this study reflected very low levels of ultraviolet light. The data underlying this figure can be found at https://doi.org/10.5061/dryad.j0zpc86nf [99]. PCA, principal component analysis; UMAP, uniform manifold approximation and projection.

of Holland's birds-of-paradise separately (Fig 5F) reveals that the hybrid's colors are—in general—intermediate between those of the parent species. Closer examination of individual patches sheds light on this. The breast color of the King and Magnificent is green, with the hybrid intermediate (in terms of λmax, the wavelength of peak reflectance) between these greens (Fig 5B, 5G, and 5K). The difference is subtle—in fact the green colors are likely to be indistinguishable from one another to a bird eye (see next section)—but after inspecting the spectral shape of the typical breast colors for each of King, Magnificent, and hybrid (Fig 5K), it is clear that the hybrid green has λmax intermediate to those of the parent species. Interpreting the magnitudes of the peak reflectance for the breast patch across the specimens is more difficult. This is because some of the hyperspectral images may by chance contain more samples that are geometrically closer to the angle of peak reflectance for this iridescent patch (but see Discussion). The belly/vent color of the King is white, while the belly/vent color of the Magnificent is brown (Fig 5C and 5H). The hybrid's belly/vent is brown, more closely resembling that of the Magnificent (Fig 5H). This is evidenced by the spectra for the most typical colors (Fig 5L), with the hybrid's light brown more similar in shape to the dark brown of the Magnificent. The shoulder color of the King is red/orange, while the shoulder color of the Magnificent is yellow (Fig 5D). The hybrid's shoulder color is orange and more closely resembles the red/orange color of the King (Fig 5D and 5I), but is spectrally intermediate to the parent species.

The back of the King is red/orange, while the back of the Magnificent is yellow (Fig 5E and 5J). The hybrid's back is intermediate, with an orange-red color (Fig 5J), but it more closely resembles the King bird-of-paradise, a result that is also clear when examining the spectra for typical colors (Fig 5N).

Our results show that the degree to which the hybrid's color is intermediate between its parent species can differ among patches, and intermediate phenotypes can result in patches colored by both pigmentary and structural mechanisms. Our approach also highlights the remarkable color diversity within these generally "uniformly colored" patches—information that is overlooked using spectrophotometry. For example, by inspecting the contributions of the King, Magnificent, and hybrid King of Holland's birds-of-paradise (Fig 5G–5J) to the color distributions (Fig 5B–5E) for each patch, we see that in all cases patches contain a variety of colors, such that only one or a small number of spectrophotometric readings would likely inadequately describe the variety of spectral shapes present.

An important disclaimer is that these inferences are based on spectral shape only. They emerge from the ways in which these spectra are embedded in a visual system-independent space with relation only to physical properties of the spectra—not to animal color vision. Therefore, many of the differences we have described in color may not be perceptually meaningful, though they may hint at important mechanistic or developmental color production processes in the hybrid and its parents. To consider perceptually relevant questions in the context of signaling, we turn to visual system-dependent approaches in the next section.

**(ii) Analyzing plumage colors with a visual system-dependent approach.** Fig 6 shows sampled spectra from King, Magnificent, and hybrid King of Holland's patches embedded in

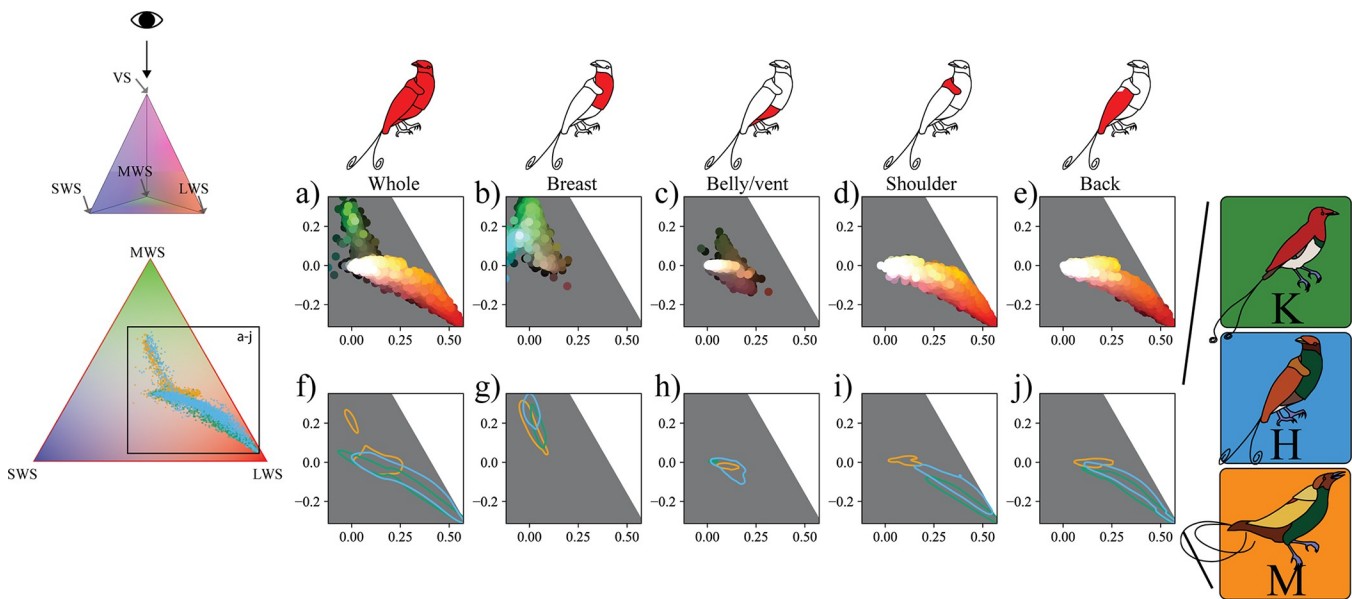

**Fig 6. Tetrahedral avian color space embedding of sampled spectra for each patch.** Top row: Icons indicate the patch represented in each column of the figure. Left top: 3D tetrahedron with icon indicating the view shown in all other panels; where we view the 3D tetrahedron down through the VS axis since none of the sampled colors reflect significant ultraviolet/violet light. Left bottom: 2D plot of embedded samples colored by species (according to key on far right) plotted in avian tetrahedral color space. The location of points indicates their relative stimulation of the photoreceptors in bird-of-paradise eyes (top: MWS, left: SWS, right: LWS). The black rectangle indicates the expanded section in each of the other panels. (a–e) Sampled spectra plotted for each patch colored according to their appearance (to human eyes, brightened by 300% for plotting) in the expanded section of the avian tetrahedral color space. (f–j) Kernel density contour plots of the same data, with contours (80% density) indicating the contribution of each species to the distribution of spectra in the color space. Line colors correspond to the different species as indicated in the key at the right of the figure. The data underlying this figure can be found at https://doi. org/10.5061/dryad.j0zpc86nf [99]. LWS, long wavelength-sensitive; MWS medium wavelength-sensitive; SWS, short wavelength-sensitive; VS, violet-sensitive.

tetrahedral avian color space (plotted in 2D for convenience), representative of a bird-of-paradise with a VS color vision system [41]. In the tetrahedral color space, an embedded point's proximity to each of the four vertices corresponds to the degree of stimulation of each of the four color cones in typical avian eyes [68]. Thus, a color that we predict would appear green to a bird would fall near the MWS vertex of the tetrahedron. However, the space accounts only for the hue and saturation of colors, not perceived brightness (indeed, black and white fall in the same location, at the center of the tetrahedron). Colors located farther apart in this space are more different in hue and/or saturation according to models of avian vision, but the space is not perceptually uniform (i.e., the same distance between points in different parts of the space do not correspond with equal discriminability [15], see Fig Q in S1 Appendix for a plot using the receptor noise limited color space). As in Fig 5, we show the distribution of colors in the upper row of Fig 6 (Fig 6A–6E). Here, an expanded section of the tetrahedral color space is shown, with points plotted for each sample colored according to their appearance (to human eyes). Panels Fig 6F–6J highlight the contribution of the King, Magnificent, and hybrid King of Holland's birds-of-paradise to the distribution of colors in the color space. The contour lines for each species enclose their most typical colors. Here, we have used an avian tetrahedral color space model, but—as with reflectance spectra obtained from spectrophotometry or images obtained from multispectral photography—the spectral data can be modeled for a great variety of animal visual systems for which the photoreceptor sensitivities are known [2,15,70].

The overall colors of the King, Magnificent, and hybrid are primarily constrained to the portions of tetrahedral color space between the achromatic point (center) and the MWS vertex, and between the achromatic point and the LWS vertex (Fig 6 and 6A). This is expected given that the predominant plumage colors are green, yellow, reddish-brown, and white. The breast colors of the King, Magnificent, and hybrid are largely overlapping in tetrahedral color space (Fig 6B and 6G) and the same is generally true for the belly/vent colors (Fig 6C and 6H; recall that hue and saturation but not brightness are encoded in this space, so buff white and brown can fall in similar locations of color space). For the shoulder and back patches, the King bird-of-paradise occupies areas of tetrahedral space that are more saturated, and slightly redder in hue than the hybrid, and the Magnificent bird-of-paradise occupies a more tightly concentrated area of yellow hue (Fig 6D, 6E, 6I, and 6J). In general, embedded points are more concentrated and overlapping than in the visual system-independent approach, making patterns of occupancy for the breast and belly/vent patches much harder to distinguish.

***Some King of Holland's bird-of-paradise color patches that are spectrally intermediate would be indistinguishable from those of its parent species when modeled with avian color vision using only one spectrum per patch.*** In order to explore whether differences between King, Magnificent, and hybrid King of Holland's plumage patches may be perceptually relevant to avian viewers, we applied the Vorobyev and Osorio [73] receptor noise model to generate chromatic contrasts (dS) and we generated luminance contrasts (dL, implemented in [70]). These contrasts compare two colors in terms of hue/saturation and brightness respectively, and—in general—contrast values above one are expected to be discriminable by a given visual system [76]. A caveat of this approach is that a number of important parameters (cone density ratios, Weber fractions) must be assumed, the choice of which strongly influences calculated contrasts [76]. Due to these assumptions, and because the models assume discrimination at the limit of physiological ability under ideal conditions, more conservative threshold contrast values of two to four may be more appropriate when determining whether two colors are discriminable [77,78]. For simplicity, we calculated contrasts using the median spectrum for each patch on each specimen, and we considered two colors to be discriminable if the contrast value was $\geq 3$ (see Fig I in S1 Appendix). The breast patch is not likely to be discriminable (in

terms of chromatic or luminance contrast) among the King, Magnificent, and hybrid birds-of-paradise. The belly/vent patch is likely to be discriminable between the hybrid and King, and between the Magnificent and King, on the basis of luminance contrasts. However, the belly/vent patch is likely to be indiscriminable between the hybrid and Magnificent. For the shoulder and back chromatic contrasts, the King and hybrid are likely discriminable from the Magnificent, but unlikely to be discriminable from one another. On the basis of luminance contrasts, the shoulder and back patches of the hybrid are likely discriminable from both the King and Magnificent (see Fig I in S1 Appendix).

While a visual system-independent approach uncovered some differences between colors, for instance in breast coloration, visual models indicate that not all of these differences are perceptually relevant (when assessed using a single spectrum for each patch, but see Fig Q in S1 Appendix). This analysis highlights the importance of tailoring the approach (visual system-independent versus visual system-dependent) to the scientific question at hand. If we were interested in the evolution of pigmentation in the context of signaling behavior, a visual system-independent approach may produce irrelevant or even misleading results. In contrast, if we were interested in the physical or optical properties of pigmentation irrespective of visual system, the situation would be reversed.

**(iii) Analyzing the color of bird-of-paradise tail feathers.** Many birds-of-paradise have unusual tail feathers. The King, Magnificent, and hybrid King of Holland's birds-of-paradise each have a modified central pair of iridescent tail feather "wires" that appear to have barbs only on the outside edge [40]. These modified tail wires are variably curved; the tail of the King bird-of-paradise resembles a tightly wound green spatula or disk, while the tail of the Magnificent bird-of-paradise is sickle-shaped and teal (Fig 7A). The hybrid bird-of-paradise tail appears to be intermediate to these in both color and shape (Fig 7A). Using quantitative methods, we explored the extent to which hybrid tail color is intermediate to the parent species (Fig 7B–7D). We extracted samples from hyperspectral images of the dorsal side of the tails of each species (approximately 500 samples per tail feather per specimen). We then calculated PCA coordinates for all samples before using UMAP to embed the spectral distances in a visual system-independent color space (Fig 7B and 7C). The aquamarine color of the barbs of the tail feathers in the hybrid King of Holland's bird-of-paradise appears to be intermediate to the

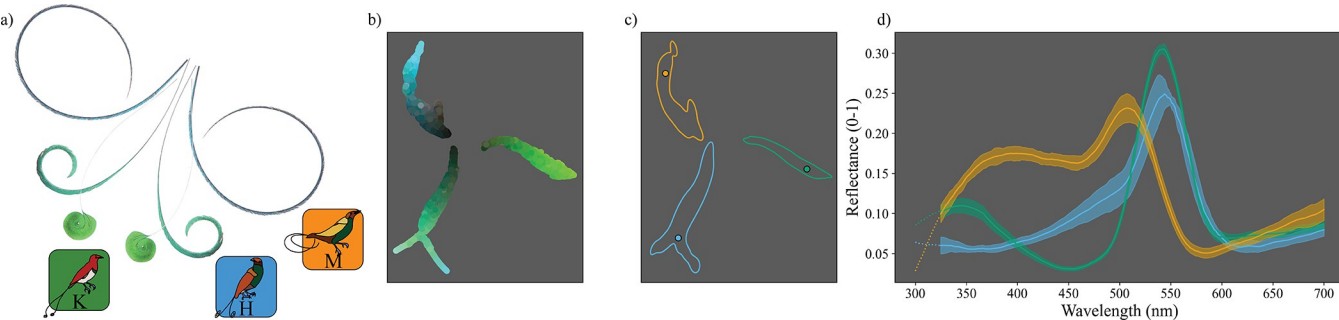

**Fig 7. Analyzing the shapes and colors of bird-of-paradise tails.** The modified tail feathers (tail wires) of the hybrid King of Holland's bird-of-paradise appear to be morphologically intermediate to those of the King and Magnificent birds-of-paradise. (a) Overlaid photographs of the dorsal side of the central modified tail feathers from each of the species, as indicated by the location of the species icons. Photos: David Ocampo. (b) A plot of a UMAP embedding of PCA coordinates for spectra sampled from the pennaceous parts of each species' tails colored according to their appearance (to human eyes, brightened by 300% for plotting). (c) Kernel density estimates (80%) showing the contribution of each species to the color space shown in panel (b). Colors indicate species as per the species icons (a). Points indicate manually selected points of interest for plotting in panel (d). (d) Plots of the reflectance of the closest 20 spectra to each point of interest shown in panel (c). Bold lines indicate the median, shaded area indicates the median absolute deviation, and color indicates species. The data underlying this figure can be found at https://doi.org/10.5061/dryad.j0zpc86nf [99]. PCA, principal component analysis; UMAP, uniform manifold approximation and projection.

emerald green of the King bird-of-paradise and the teal shade of the Magnificent bird-of-para-dise. This observation was supported by sampling and embedding using the visual space-inde-pendent methods described above, and inspection of the sampled spectra (Fig 7B and 7C). The reflectance of Magnificent bird-of-paradise tails shows an ultraviolet-blue spectrum, while the King bird-of-paradise shows a saturated green, with the hybrid appearing to be a mix of the two (Fig 7D). We also explored the striking shape of the tail feathers and whether a simple model could replicate the curvature of the rachis of each species. We found that a simple model of bias in the growth of sections of the rachis during feather growth replicated the tail shapes, and that the hybrid has bias generally intermediate to the King and Magnificent birds-of-paradise (see Fig U in S1 Appendix).

## Discussion

The hyperspectral imaging pipeline we presented here provides the tools necessary to harness the power of off-the-shelf hyperspectral imagers in animal color research. We have demon-strated how hyperspectral imagers can overcome the spatial limitations of spectrophotometers and the spectral limitations of multispectral photography to provide a versatile and powerful tool for quantifying animal coloration. We applied visual system-independent and visual sys-tem-dependent methods to quantify the diversity of plumage colors present in a rare hybrid and its parent species—and to explore the extent to which the hybrid's plumage colors are intermediate composites of those of its parent species.

The plumage colors of the hybrid are generally intermediate to those of the King and Mag-nificent birds-of-paradise (Figs 5A, 5F, 6A and 6F). However, the degree of intermediacy appears to differ by patch. The back and shoulder of the hybrid are more similar to the King bird-of-paradise (Fig 5D, 5E, 5I and 5J), whereas the belly/vent is more similar to the Magnifi-cent bird-of-paradise (Fig 5C and 5H). In both the breast and tail, we found that the hybrid was strikingly intermediate: this is most clear when inspecting the spectral shapes of the reflec-tance of tail feathers (Fig 7D). The hybrid's plumage patches span the gamut of avian color production from carotenoid and melanin pigments to structural coloration, so differences in intermediacy among patches may reflect different genetic and developmental effects of hybrid-ization across coloration mechanisms. By applying avian visual models, we show that some but not all of the differences in color among the hybrid, King, and Magnificent birds-of-paradise are likely to be perceptually discriminable to birds and therefore potentially behaviorally rele-vant (see Fig I in S1 Appendix, at least when modeled using only a single spectrum per patch, also see Fig Q in S1 Appendix). This observation highlights the importance of tailoring the approach to the scientific question (i.e., modeling with respect to visual signal receiver or not), which is possible even after data collection due to the versatility of hyperspectral images. Fur-ther, we found that in general, patches are not homogenous: summarizing plumage using sin-gle or average color spectra may obscure substantial differences within patches. Overall, we showed that hyperspectral imaging is a feasible, efficient, and effective method for collecting detailed color information from museum specimens. Animal color research need not be lim-ited by the trade-off between spectral and spatial resolution, and hyperspectral imaging repre-sents a promising avenue for future researchers.

### Hyperspectral imaging: Future challenges and opportunities

Moving forward, several challenges and opportunities remain for developing methods for hyperspectral imaging of animal coloration. Angle-dependent (iridescent or any non-Lamber-tian) colors are difficult to quantify [79]. For these colors, the angle of observation, the angle of the specimen's surface (surface normal), and the angle/s of incident illumination (together

termed the viewing geometry) determine in large part the perceived (and measured) color [74]. In spectrophotometry, this problem is often approached either by integrating across all possible viewing geometries (using an integrating sphere) or by systematically taking many measurements using various viewing geometries [80]. Generally, in multispectral imaging (and in the current paper, with hyperspectral imaging) this problem is approached by diversifying the viewing geometries used in color measurement by taking multiple measurements of the same patch (for instance, by imaging the sample at two or more angles e.g., [41,81]). An extension of this approach is to more precisely understand the viewing geometries within each image. Any image of a 3D (non-flat) iridescent or angle-dependent colored sample will contain a variety of viewing geometries because the surface normal of the sample is not uniform. In this paper, we piloted a method to recover information about the viewing geometries contained within an image through registration with a separately generated 3D model. In this way, we supplemented our hyperspectral images with surface normal information to better explore the angle dependency of the breast patch in the three birds-of-paradise (see Fig 4). This approach could easily be adapted for use with multispectral images. A remaining goal is to better control, diversify, and record the direction of illumination for color measurements in hyperspectral images of animal colors. Kim and colleagues [37] introduced a related approach: the authors collected 3D surface information concurrent with hyperspectral imaging, allowing registration between the 3D structure of a bird specimen while color measurements were taken. However, this approach required specialized and highly customized equipment [37]. Further development of these, or similar, approaches will allow animal color researchers to move away from methods that implicitly assume that all animal colors are Lambertian (perfectly diffusely reflective) and to thereby assess animal colors in a more realistic way.

Animal coloration is often patchy (i.e., a given animal's color is distributed across the body in patches), which complicates analysis of overall color difference and similarity between individuals or species. In general, researchers define color patches *a priori*, allowing them to compare homologous locations—an approach we took here. It is becoming possible, however, to generate patches for animal colors automatically or semi-automatically. For example, powerful clustering algorithms are being developed for quantifying animal coloration patterns [2]; other algorithms are being used to segregate a sample from the background [82,83] and to find physically homologous body locations in images of animals [84,85]. It seems likely that soon some combination of these efforts will result in a robust means by which to identify discrete patches in an objective and automated way, free of human biases.

At present, hyperspectral imaging typically has relatively poor temporal resolution (Table 1). Hyperspectral imaging of live animals moving in the wild (at least with high spectral and spatial resolution, and broad wavelength sensitivity) is not yet practical because images take minutes to generate. To some extent, this problem is also true of multispectral photography, although a study recently introduced the use of a beam splitting mirror to photograph a scene using ultraviolet- and visible light-sensitive cameras simultaneously (typically at framerates up to 30 per second), allowing the collection of multispectral video [7]. Both hyperspectral imaging and multispectral photography may benefit from better integration of imaging with 3D modeling and animation, where dynamic animal motion may be introduced and investigated *in silico* ([24,86]). For now, hyperspectral imaging for animal coloration as implemented here may be largely limited to museum specimens. However, museum specimens do not always perfectly represent the colors of live animals. For example, the physical posture for the specimen chosen during preparation can strongly influence the colors visible on the animal surface. Further, as specimens age, their color can change or fade, especially if the specimen is exposed to sunlight [3]. Lastly, complete metadata is often unavailable for historical specimens, sometimes limiting their use for study. Overall, hyperspectral imaging is currently primed for

imaging animal specimens, but *in situ* animal color measurements will have to wait for advances in hardware (such as video-speed capture) to allow rapid collection of full-frame hyperspectral data with high spatial and spectral resolution and broad wavelength sensitivity.

Although our study focused on plumage colors, hyperspectral imaging holds great promise for the study of animal color in diverse invertebrate and vertebrate taxa, including butterflies [28,34], beetles [30], spiders [29], and cephalopods [32], to name a few. Hyperspectral imaging is likely to yield new insights about the evolution of color and pattern in bird eggs, bird nests, fruit, and flowers—and could reveal rich details about animal color in the contexts of urbanization and climate change. Similar insights might be gleaned from spectrophotometry and multispectral imaging, too—but only hyperspectral imaging provides high-resolution spectral and spatial data simultaneously. Hyperspectral imaging systems remain expensive. Resonon is currently selling its ultraviolet-sensitive imaging system for about $55,000 USD, and the light source and lens used here increase the total cost to around $75,000 USD. However, we expect that costs will continue to decrease in the future. As hyperspectral imaging becomes increasingly mainstream in animal coloration research, we hope that researchers will collaborate to establish shared tools, software, and "best practices," all of which could help to fuel large-scale efforts to digitize and characterize diverse animal specimens. Future software should ideally be freely available, highly computationally efficient (and amenable to cloud computing), and inter-operable with existing tools for multispectral image analysis (such as MICA/QCPA [2,5]) and spectral analysis (such as the R package pavo [70]).

## Insights into the plumage color of a hybrid bird-of-paradise

We found evidence of intermediacy (in terms of spectral properties) across patches of color generated by both pigmentary (back and shoulder, Fig 5M and 5N) and structural (breast and tail, Figs 5K and 7D) mechanisms in the hybrid. In birds, yellow and red colors are typically generated by carotenoids, which are metabolized from dietary sources for deposition in the integument [87,88]. Research has shown that intermediate orange pigmentation in the hybrid offspring of red and yellow birds can correlate with greater concentrations of incompletely oxidized carotenoid products [89–92]. In addition, the genetic basis of carotenoid coloration in birds is becoming clearer [87,88,93–95]. Future work might explore intermediacy in carotenoid coloration in the hybrid bird-of-paradise through characterization of the carotenoid composition of their feathers as well as investigation of hybrid genetics at relevant loci.

We also described intermediacy in the color of the hybrid's structurally colored breast and tail patches (Figs 5K and 7D), a finding that contrasts with those of Barrera-Guzmán and colleagues [64] and Eliason and colleagues [65], who found —in other avian taxa— non-intermediate coloration in patches of structural coloration in hybrid offspring. Here, we found that the structural colors from both parent species instead gave rise to intermediate hybrid breast and tail colors. Some nanostructures in the Magnificent bird-of-paradise have been investigated; for example, melanosomes in the breast patch are known to be solid (as opposed to air-filled, though they appear to be porous [96], as cited in [97]), round and rod-like, about 130 nm in diameter, and arranged in double layers apparently unique to Paradisaeidae [97]. Less is known about the melanosomes or their arrangement in the King bird-of-paradise, though there are hints that its melanosomes are also round and solid [97]. Whether and which nano- or micro-structural parameters of the hybrid breast and tail patches are intermediate to that of the parents could be investigated directly using microscopy.

Mysteries about the hybrid King of Holland's bird-of-paradise remain. For example, the Magnificent bird-of-paradise is strongly fluorescent under a blacklight (pers. comm. Glenn Seeholzer). We confirmed this: fluorescence is especially prominent in the highly reflective

yellow shoulder/cape patch (see Fig V in S1 Appendix). Neither the King nor the hybrid bird-of-paradise shows significant fluorescence: whatever the mechanism, we do not see evidence of intermediacy in the hybrid. We note that strong fluorescence can complicate the measurement of animal color; the light produced by fluorescence can introduce errors in inferred reflectance (pers. comm. Jolyon Troscianko). Exactly what mechanism generates fluorescence in the Magnificent bird-of-paradise, and why it appears absent in the hybrid, could be investigated by sectioning and imaging feather samples—and by testing for any influence of specimen preparation methods [98]. Further characterization of the fluorescence, including specifics of the excitation and emission wavelengths, might also be studied using a hyperspectral imager in concert with spectral filters and/or a monochromator. Another question relates to bare-part coloration across these birds. One disadvantage of avian museum specimens is that bare-skin colored parts do not retain their colors from life. Both the King and Magnificent birds-of-paradise have dark blue legs in life [40]. Additionally, the inside of the mouth and the tongue of the Magnificent bird-of-paradise are a striking pale green, while the inside of the King's mouth is pale aqua-green [40]. Since the hybrid is only known from specimens, how and whether these bare-part colors compare may never be known unless live hybrids can be found.

In summary, we have shown that hyperspectral imaging is becoming a practical and advantageous alternative to spectrophotometry and multispectral photography for measuring animal colors. Hyperspectral images have high spectral and spatial resolution, allowing for detailed visual system-independent and visual system-dependent analyses of animal colors. Using the example of the birds-of-paradise, we provide a guide for capturing and analyzing hyperspectral data, which we hope other researchers will adopt and update. We also provide example code showing how hyperspectral data can be easily incorporated into existing spectra- and image-based color analysis toolkits (pavo: [70], MICA/QCPA: [2,5]). Overall, hyperspectral imaging may help galvanize high-throughput projects that allow for a more complete description of animal color, thereby providing new insights into the mechanistic, evolutionary, and developmental processes that drive phenotypic diversity.

## Supporting information

**S1 Appendix. Supplementary text and figures (Tables A and B and Figs A–V).** Raw data and code for all supplementary figures are available (see the Data availability section) (.PDF). (PDF)

**S1 Video. Video illustrating 3D models of all specimens (2 King bird-of-paradise, 2 Magnificent bird-of-paradise, and 2 hybrid King of Holland's bird-of-paradise).** In the first half of the video, the models are colored according to the RGB images used to generate the models (see main text and S1 Appendix). In the second half of the video, models are colored according to the surface normals of the visible polygons in the video. Polygons facing left relative to the camera are teal, while those facing right are magenta. Polygons facing up relative to the camera are darker, while those facing down are lighter. Models are to scale. The sphere in the bottom left has diameter 2 cm, and the cube in the bottom right has edge length 2 cm. (.MP4). (MP4)

## Acknowledgments

We thank colleagues at Resonon for detailed technical discussions about hyperspectral imaging and for sharing equipment. We are grateful to the American Museum of Natural History (New York City) for loaning us the birds-of-paradise specimens; we especially thank Paul Sweet, Joel Cracraft, Brian Smith, Tom Trombone, and Bentley Bird. We appreciated

discussions with Glenn Seeholzer, who observed fluorescence in the plumage of the Magnificent bird-of-paradise. We thank David Ocampo for photographing the specimens, Derya Akkaynak, Rick Prum, Roger Hanlon, Innes Cuthill, and Henry Knowles for early discussions about hyperspectral imaging, and Audrey Miller, Klara Nordén, Jarome Ali, Sarah Solie, Harold Eyster, Rosalyn Price-Waldman, and Lazarena Lazarova for constructive feedback.

## Author Contributions

**Conceptualization:** Benedict G. Hogan, Mary Caswell Stoddard.

**Data curation:** Benedict G. Hogan.

**Formal analysis:** Benedict G. Hogan.

**Funding acquisition:** Mary Caswell Stoddard.

**Investigation:** Benedict G. Hogan, Mary Caswell Stoddard.

**Methodology:** Benedict G. Hogan, Mary Caswell Stoddard.

**Project administration:** Benedict G. Hogan, Mary Caswell Stoddard.

**Resources:** Mary Caswell Stoddard.

**Software:** Benedict G. Hogan.

**Supervision:** Mary Caswell Stoddard.

**Validation:** Benedict G. Hogan, Mary Caswell Stoddard.

**Visualization:** Benedict G. Hogan, Mary Caswell Stoddard.

**Writing – original draft:** Benedict G. Hogan, Mary Caswell Stoddard.

**Writing – review & editing:** Benedict G. Hogan, Mary Caswell Stoddard.

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
