## [Editor Report · Decision Letter 0]

21 Dec 2023

Dear Dr Hogan, 

Thank you for submitting your manuscript entitled "Hyperspectral imaging is a powerful tool for animal coloration research: A case study of a rare hybrid bird-of-paradise" for consideration as a Methods and Resources Article by PLOS Biology. Please accept my apologies for the delay in getting back to you as we consulted with an academic editor about your submission. 

Your manuscript has now been evaluated by the PLOS Biology editorial staff, as well as by an academic editor with relevant expertise, and I am writing to let you know that we would like to send your submission out for external peer review.

Once your full submission is complete, your paper will undergo a series of checks in preparation for peer review. After your manuscript has passed the checks it will be sent out for review. To provide the metadata for your submission, please Login to Editorial Manager (https://www.editorialmanager.com/pbiology) within two working days, i.e. by Dec 23 2023 11:59PM.

Kind regards,

Richard

Richard Hodge, PhD

rhodge@plos.org

PLOS

---

## [Decision Letter · Decision Letter 1]

21 Feb 2024

Dear Dr Hogan,

Thank you for your continued patience while your manuscript "Hyperspectral imaging is a powerful tool for animal coloration research: A case study of a rare hybrid bird-of-paradise" was peer-reviewed at PLOS Biology as a Methods and Resources Article. Please accept my sincere apologies for the delays that you have experienced during the peer review process. Your manuscript has now been evaluated by the PLOS Biology editors, an Academic Editor with relevant expertise, and by three independent reviewers. Please note that Reviewer #2 has provided the review as a PDF document which is attached to this decision letter. 

In light of the reviews, which you will find at the end of this email, we would like to invite you to revise the work to thoroughly address the reviewers' reports.

As you will see, the reviewers are generally very positive about your hyperspectral imaging pipeline and think the method will be useful for the field. However, Reviewers #2 and #3 raise some overlapping concerns, including the overall framing of the paper and noting that a greater emphasis should be placed on the methodological advance. In addition, Reviewer #3 notes that the manuscript lacks direct benchmarking to other existing methods in order to demonstrate a proven advantage. After discussions with the Academic Editor, we agree that this would be a valuable addition and we ask that direct benchmarking experiments are included in the revision. Finally, Reviewer #2 asks that additional reporting details should be included for the lighting and elaborating on other applications offered by the approach.

Given the extent of revision needed, we cannot make a decision about publication until we have seen the revised manuscript and your response to the reviewers' comments. Your revised manuscript is likely to be sent for further evaluation by all or a subset of the reviewers.

**IMPORTANT - SUBMITTING YOUR REVISION**

*Re-submission Checklist*

*Published Peer Review*

*PLOS Data Policy*

*Blot and Gel Data Policy*

Sincerely,

Richard

Richard Hodge, PhD

rhodge@plos.org

REVIEWS:

Reviewer #1: GENERAL

A splendid paper outlining a potentially transformative new approach to avian color quantification. I want to highlight, in particular, the beautiful and informative figures. The authors strike a nice balance among justifying this new approach relative to two others, describing the pipeline and potential applications, and illustrating its use with an interesting example. I have mainly minor comments throughout, see below for details.

No methods section? Can methods and results be separated? Not sure about journal format, but L233-455 read like methods and L456-678 read like results.

Figure 2: Put hybrid image between parentals, logical and aligns better with subsequent figures.

RESULTS

L242: Double-check your figure panel letters here.

L261: Commentary about missing those shortest 25 nm here or elsewhere? These species don't have much UV reflectance, but how could this affect species with more prominent UV patches? Might be interesting to do an analysis validating the extrapolation technique.

L457: This section is somewhat redundant with some previous sections. I think you've already clearly made the case for hyperspectral imaging and how/why it's different/better than the other two methods (e.g., Table 1).

Figure 5: This is the only figure with embedded descriptive text, could be removed and covered in the caption.

L495: I would rethink the use of the word "favors" here. "Resembles" might do.

L552: I think "further" should be "farther" here, as this refers to an actual and not metaphorical distance. Look for other instances of this throughout the ms if there are any.

DISCUSSION

L742: I think this section could be expanded upon to discuss the potential limitations of basing color measurements/comparisons off specimens alone (e.g., effects of age, preservation method/quality, metadata availability).

L753: Watch for capitalization consistency in species names.

Reviewer #2 (Daniel Hanley, signs review): Please see the PDF document attached to this e-mail for the full review.

Reviewer #3 (Jolyon Troscianko, signs review): This paper showcases a pipeline for hyperspectral imaging of museum specimens, and uses it to investigate the provenance of a putative hybrid bird of paradise.

Overall I really like the methods, and wholeheartedly welcome the release of open-source tools for the community. The manuscript is nicely written, though I feel it could be shortened in parts. Hyperspectral imaging techniques will be of increasing interest to a wide audience, and birds-of-paradise are a charismatic group to investigate.

I make a large number of minor points below, but none should be difficult to deal with, it mostly reflects the fact that (I think) I understand the methods quite well.

I have just two main concerns (again, easy to address, but might entail some work). One trying to work out the rationale for this paper, the other technical.

Rationale: The core hypothesis being tested is not entirely convincing as-is. e.g. deciding whether a species of apparently intermediate coloration/physiology between two other species really represents a hybrid would ideally need a full phylogenetic comparison (measure lots of species with this technique, including other known hybrids), or at least some meaningful out-group, control or reference. As such, there's no clear alternate hypothesis. For example, the statement that the (alleged) hybrid is 'strikingly intermediate' has no baseline for comparison. I don't know how much the authors want to sell this finding as important though, as I feel the real value of the paper lies in its methodology (which is great), and this hypothesis is just used as a demo. In any case it's shortcomings are worth discussing. However, if the authors do want to sell the methodology there are some important things I'd like to see to convince readers that this method is worth using themselves. e.g. How does it compare to other methods in practice (show evidence)? Would alternative conclusions be reached if this study if other (cheaper/easier) methods were used (i.e. direct comparison of the output of methods, or modelling of circumstances where high spectral resolution is required?)

Processing methods: (L342) I'd like more information on the rationale for the sampling and smoothing applied here. And importantly I'd like to see some raw data before any smoothing is applied. 45 nm is a surprisingly spectrally wide window, and 1nm a surprisingly high spectral resolution. This stage will be computationally inefficient as it's artificially increasing the spectral resolution from that of the native hardware. This will also cause all following calculations to be equally inefficient for no apparent gain in accuracy. Up-sampling might give false impressions of spectral resolution. Why not just downsample the entire cube to e.g. a 10nm resolution that wouldn't need smoothing, and would give pretty similar effects for less computational overheads? Or even more elegant, just work with the spectral range you're given by the camera and adjust the bin-width in the calculations as necessary (as these will presumably be non-linear).

Related question: L336 It strikes me that the processing pipeline would be far more efficient if each pixel were first converted to avian cone-catch values (or non-system-specific bins), and then the rest of the analysis done simply on conventional images. This doesn't strike me as particularly computationally tricky or slow. Even a single processor should take less than a minute to churn through a 5MPx200 channel cube with efficient code (python might not be great for this heavy lifting though). I haven't seen the code, but would like to have a look,

Minor points: 

It's worth pointing out that high spectral resolution is not always required. Natural reflectance spectra and solar illumination rarely have sharp changes across a 5-10nm range. This means we can infer the way colours will look to a given animal under given (natural) lighting with a fairly high degree of confidence. The reflectance profiles shown in fig.1 for example, would behave very predictably at lower spectral resolution sampling. Where this fails is when either the light source is non broadband (important for my current work in artificial light pollution), or when reflectance profiles are particularly rare (e.g. sometimes achieved in structural coloration).

L20 I'd query whether hyperspectral data are computationally costly. It depends on the spectral, spatial, and temporal resolution, but the hardware has typically been the limiting factor here. I also query the affordability below.

Table 1. I'd like a further row of "temporal resolution" added. This highlights the issue with using multispectral and hyperspectral cameras and moving targets. e.g. some systems can work instantaneously, or capture video, but others require slower scanning/repeat photos taking. The camera used here requires scanning, so cannot take instantaneous images. It's perhaps worth referencing Vasas et al's recent dichroic camera PLoS Biol paper here as an example of low-spectral, but high spatial and temporal resolution (you cite the pre-print, but can cite the full thing now).

L77 multispectral imaging doesn't always use one camera spectral channel for one receptor channel (though often does). Worth just rephrasing slightly. e.g. people often model achromatic and dichromatic vision using RGB cameras.

L91 Multispectral and hyperspectral data are both able to generate reflectance spectra. It's simply a matter of resolution and range. Indeed, conventional spectrometry can't handle the full UV-vis-IR-thermalIR ranges required to understand all the effects discussed here.

L104 'brightness' is not the right word here. The system will be measuring radiance (specifically energy of light in a known solid angle or a known spectral range).

L141 Could you say how much the system used here cost? This would be valuable for readers. Include the broadband stable lighting setup used (these can be super expensive too), plus the ~$10,000 UV-VIS lens. It'll all add up quite a bit. Presumably $20-30k? Which dents the angle that this is affordable.

L254 Personally I'd like to see other colour spaces added. e.g. the RNL Chromaticity colour space is much easier to use, interpret, and analyse with reference to discriminablility (though I'd caution that almost all colour spaces are pretty flawed, so it's not worth getting too fussy).

L247 Are you getting at measuring gloss and iridescence here? i.e. goniometry? If so, maybe be explicit. You bring this up later at least, but generally there's scope of more concise writing.

L264 This spatial correction is interesting, presumably normalising each pixel to account for non-diffuse lighting. But this highlights real problems for scanning/imaging anything that's not flat. Things nearer the lights than the 2D white calibration tile will appear artificially lighter. So for 3D subjects illuminated from non-diffuse, close light sources care needs to be taken for accurate reflectance estimates.

L262 birds have very low sensitivity to UV below 325nm, and the sun doesn't give out much energy at these wavelengths either. I'm just saying this in case anyone worries that part of the range is 'missing' - it's not likely to be an issue as far as I can tell.

L288 Really happy to see use of Python (great to keep code open source). But is it all in python? As an interpreted language it's not always very quick for image processing.

L355 I disagree here - cameras are frequently used to compare spectral reflectance. e.g. NDVI methods, and enormous numbers of methods and studies. I get what the authors are hinting at, but the only real difference with the system used here is its spectral resolution, not the ability of the system to measure spectral reflectance across a given spectral range. Indeed, even common light meters (not cameras) often work by comparing a 'full-spectrum' to an infrared measurement to determine brightness in the visible range.

L376 Why not just focus on the PCA-based method then? I like the PCA method, and it's used a fair bit in the field (as you say, it's easy to understand and apply stats etc...). Could make the paper more concise, and help the readers.

L408 But you are only measuring each point once or twice without actively controlling the angles, so unlike goniometry you can't quite work out iridescence with this method (without more work).

L535 How sure can we be that these differences aren't due to noise of some sort? Also, all spectral recordings will differ, but how can we say different any two are without some point of reference or threshold?

L590 If you're converting to RNL values why use the tetrahedral colour space at all? Just convert to RNL XYZ chromaticity, then you have your hue axes just like the tetrahedral space, but now Euclidean distance is equal to RNL delta-S values (i.e. perceptually uniform-ish). Hue and discriminability can be compared in one go.

L595 What visual system (spectral sensitivity curves, Weber fraction and cone ratios do you use here?)

L628 I like this method for modelling feather growth. Though maybe you could have just fitted a bezier spline? It might be worth mentioning why developmentally-inspired methods are used rather than just absolute modelling here, but this approach is not used for spectral reflectance curves (even though the discussion goes over many developmentally relevant colour creation mechanisms)?Playing devil's advocate, this is a lot of work and effort to say something that is obvious to any viewer of the feathers (the intermediate one is intermediate). I really like the methods, but their use doesn't seem enormously well justified here - persuade me I'm wrong.

L703 As above, I'm not convinced there are meaningful differences shown between spectra of the same patch. What exactly would 'meaningful' be given everything in the world has some degree of difference?

L798 Fluorescence is a pig! Relating to my above point, artificial light sources aren't able to simulate how something looks if strong fluorescence is involved as it violates the illuminant/reflectance equations. Few people realise this, so it's perhaps worth pointing out to readers. Your use of xenon lighting was fairly good in this regard, and moderately likely to replicate natural viewing (although the UV component will be far higher than sunlight). Most other light sources will create more errors in inferred reflectance.

The discussion gets very speculative regarding bird of paradise crosses, and to be honest I don't find it particularly useful if the thrust of this paper is methods. The bird of paradise based work here doesn't shed light on many of the issues touched on in the discussion.

Jolyon Troscianko

---

## [Decision Letter · Decision Letter 2]

13 Aug 2024

Dear Dr Hogan,

Thank you for your patience while we considered your revised manuscript "Hyperspectral imaging is a powerful tool for animal coloration research: A case study of a rare hybrid bird-of-paradise" for publication as a Methods and Resources Article at PLOS Biology. This revised version of your manuscript has been evaluated by the PLOS Biology editors, the Academic Editor and the original reviewers. Please note that Reviewer #2 has provided their review as a PDF file which is attached to this letter.

Based on the reviews, I am pleased to say that we are likely to accept this manuscript for publication, provided you satisfactorily address the remaining minor points raised by the reviewers. Please also make sure to address the following data and other policy-related requests that I have provided below (A-F):

(A) We routinely suggest changes to titles to ensure maximum accessibility for a broad, non-specialist readership, and to ensure they reflect the contents of the paper. In this case, we would suggest a minor edit to the title, as follows. Please ensure you change both the manuscript file and the online submission system, as they need to match for final acceptance:

“A user-friendly pipeline for the generation and analysis of hyperspectral images for animal coloration research”

(B) You may be aware of the PLOS Data Policy, which requires that all data be made available without restriction: http://journals.plos.org/plosbiology/s/data-availability. For more information, please also see this editorial: http://dx.doi.org/10.1371/journal.pbio.1001797

-Supplementary files (e.g., excel). Please ensure that all data files are uploaded as 'Supporting Information' and are invariably referred to (in the manuscript, figure legends, and the Description field when uploading your files) using the following format verbatim: S1 Data, S2 Data, etc. Multiple panels of a single or even several figures can be included as multiple sheets in one excel file that is saved using exactly the following convention: S1_Data.xlsx (using an underscore).

-Deposition in a publicly available repository. Please also provide the accession code or a reviewer link so that we may view your data before publication. 

Figure 1C, 4F, 5A-N, 6A-J, 7B-D, S3, S4, S5A-N, S6A-N, S7, S8, S9, S16, S17, S21

(C) I tried to search for the hyperspectral image data deposited in the Dryad database using the DOI provided in the Data Availability Statement, but could not find it. I would be grateful if you could please make sure that the DOI is correct or to make the data publicly available at this stage. 

(D) Please also ensure that each of the relevant figure legends in your manuscript include information on *WHERE THE UNDERLYING DATA CAN BE FOUND*, and ensure your supplemental data file/s has a legend.

(E) Please ensure that your Data Statement in the submission system accurately describes where your data can be found and is in final format, as it will be published as written there. 

(F) Please note that we cannot accept sole deposition of code in GitHub, as this could be changed after publication. However, you can archive this version of your publicly available GitHub code to Zenodo. Once you do this, it will generate a DOI number, which you will need to provide in the Data Accessibility Statement (you are welcome to also provide the GitHub access information). See the process for doing this here: https://docs.github.com/en/repositories/archiving-a-github-repository/referencing-and-citing-content

We expect to receive your revised manuscript within three weeks. 

*Published Peer Review History*

*Press*

Best wishes,

Richard

Richard Hodge, PhD

rhodge@plos.org

Reviewer remarks:

Reviewer #1: Well I feel like a dolt for giving such a mild-mannered and brief review compared to the two incredibly detailed critiques you received from the other two reviewers! Thankfully your ms made it onto the desks of two of the foremost experts in this field (plus me!). Having read through their reviews, your responses, (and the few minor responses warranted to my comments), and your revised ms, it's clear that this version has been greatly improved. In retrospect, I agree with them that - for this type of paper - the emphasis should have been more on the technique (both in terms of greater technical detail and greater discussion of applicability) than the BoP example. Your revision accomplishes this change of focus nicely. I commend the authors on a thoughtful and nuanced revision.

Reviewer #2 (Daniel Hanley, signs review): Please see the attached PDF

Reviewer #3 (Jolyon Troscianko, signs review): Overall the authors have done a very good job at addressing my concerns, and those of the other reviewers, making a number of significant changes to the manuscript and analyses. Although the authors haven't provided new evidence that this method is superior to others (e.g. benchmarking), their discussion provides readers with enough information to make an informed judgement, and the method is a valuable addition to the field.

Minor points:

Re. the addition of temporal resolution to table 1, I would say that spectrometers should be high (they measure all the dimensions they can measure near-instantaneously). Multispectral should be high for RGB, and medium for wider spectral bands, and hyperspectral should be medium or low. Obviously, hyperspectral methods must trade off the dimensions against each-other, so will typically be slower. Otherwise, you could specify that the table only refers to techniques in the vis-UV range.

Regarding the spectral filtering, FigS2 is useful, and the raw spectra don't look bad. The opsin curves applied to these raw spectra will be smooth, and will also have the effect of smoothing out the noise substantially. In any case, the effects are now easy for the reader to see.

Regarding costs, thanks for adding these - pretty pricey! I would disagree that costs are coming down though. Off-the-shelf hyperspectral imagers simply don't have a large enough consumer base, and UV optics and light sources haven't become cheaper in over the past 15 years or so I've been paying attention. Given the renewed focus on methods, I would still recommend you add an estimate of the full cost, presumably nearer $70k?

---

## [Editor Report · Decision Letter 3]

27 Sep 2024

Dear Ben,

On behalf of my colleagues and the Academic Editor, Gail Patricelli, I am pleased to say that we can accept your manuscript for publication, provided you address any remaining formatting and reporting issues. These will be detailed in an email you should receive within 2-3 business days from our colleagues in the journal operations team; no action is required from you until then. Please note that we will not be able to formally accept your manuscript and schedule it for publication until you have completed any requested changes.

In addition, please ensure that the data deposited in the Dryad database is now made publicly available at this stage.

PRESS

Best wishes, 

Richard

Richard Hodge, PhD

rhodge@plos.org

PLOS
